# Extensive inland thinning and speed-up of Northeast Greenland Ice Stream

Shfaqat A. Khan[1✉], Youngmin Choi[2], Mathieu Morlighem[3,4], Eric Rignot[4], Veit Helm[5], Angelika Humbert[5], Jérémie Mouginot[6], Romain Millan[6], Kurt H. Kjær[7] & Anders A. Bjørk[8]

Over the past two decades, ice loss from the Greenland ice sheet (GrIS) has increased owing to enhanced surface melting and ice discharge to the ocean[1–5]. Whether continuing increased ice loss will accelerate further, and by how much, remains contentious[6–9]. A main contributor to future ice loss is the Northeast Greenland Ice Stream (NEGIS), Greenland's largest basin and a prominent feature of fast-flowing ice that reaches the interior of the GrIS[10–12]. Owing to its topographic setting, this sector is vulnerable to rapid retreat, leading to unstable conditions similar to those in the marine-based setting of ice streams in Antarctica[13–20]. Here we show that extensive speed-up and thinning triggered by frontal changes in 2012 have already propagated more than 200 km inland. We use unique global navigation satellite system (GNSS) observations, combined with surface elevation changes and surface speeds obtained from satellite data, to select the correct basal conditions to be used in ice flow numerical models, which we then use for future simulations. Our model results indicate that this marine-based sector alone will contribute 13.5–15.5 mm sea-level rise by 2100 (equivalent to the contribution of the entire ice sheet over the past 50 years) and will cause precipitous changes in the coming century. This study shows that measurements of subtle changes in the ice speed and elevation inland help to constrain numerical models of the future mass balance and higher-end projections show better agreement with observations.

The NEGIS drains about 12% of the interior GrIS through two fast-flowing marine-terminating outlet glaciers: Nioghalvfjerdsfjord Gletscher (NG) and Zachariae Isstrøm (ZI)[10–12,21–24] (Fig. 1). This region holds a 1.1-m sea-level-rise equivalent and is characterized by an exceptional fast-flowing main trunk (which is about 600 km long and 30–50 km wide) that connects the deep interior of the ice sheet to lower-lying marine-terminating outlet glaciers (Fig. 1). Understanding the coupling of the flow speed between the upper and lower sectors of the NEGIS is crucial for reliable projections of its contribution to global sea-level rise. Although previous studies showed that satellite interferometry can provide highly detailed ice-sheet-wide velocity maps (Fig. 1a), the detection of a potential acceleration in the ice flow has been limited to marginal areas[10,11]. Here we quantify acceleration and thinning in the deeper section of the NEGIS (>100 km inland) and show that these measurements are crucial for future simulations because they can better constrain the basal conditions.

Over the past decade, the NEGIS has been rapidly speeding up, but the uncertainty of the flow speeds obtained from satellite interferometry has hampered the detection of ice flow acceleration more than 100 km upstream of the terminus[10,11,25,26]. To detect how far the ice flow acceleration has propagated upstream, we use GNSS data[12,27,28] and recent improved Sentinel-1 ice velocity products[29]. We use GNSS data from three stations located between 90 and 190 km inland; these

stations recorded data from 2016 to 2019. The GNSS receivers were installed on the glacier in a configuration that followed the centre of the main trunk. Owing to deep crevasses, NEG3 was installed 10 km south-east of the main trunk. We use these GNSS stations to validate ice flow accelerations from Sentinel-1 data. We then use Sentinel-1 flow accelerations combined with satellite altimetry to fine-tune our ice flow model to improve the response of the upstream sector to changes at the terminus.

## Observations of change from 2007 to 2022

To estimate the ice flow speed, we estimate the positions of the GNSS sites using the GIPSY-OASIS software package[30]. We calculate the position of the GNSS receivers at 15-s intervals and derive mean daily velocities for each site by fitting a trend to the position estimates using east, north and up coordinates (Extended Data Fig. 2 and Methods). GNSS time series have error sources that produce temporal correlations. To take this temporally correlated (non-Gaussian) noise into account, we use the daily solutions for each station and estimate the monthly mean ice speed and associated standard deviation (Fig. 2a–c). We correct for changes in speed related to the GNSS station moving downhill. All GNSS stations show an acceleration in the surface speed, suggesting a propagation of the flow acceleration more than 190 km inland. At NEG3, we

[1]DTU Space, Technical University of Denmark, Kongens Lyngby, Denmark. [2]Jet Propulsion Laboratory, California Institute of Technology, Pasadena, CA, USA. [3]Department of Earth Sciences, Dartmouth College, Hanover, NH, USA. [4]Department of Earth System Science, University of California, Irvine, Irvine, CA, USA. [5]Glaciology Section, Alfred Wegener Institute, Bremerhaven, Germany. [6]Institut des Géosciences de l'Environnement, Université Grenoble Alpes, Grenoble, France. [7]Section for GeoGenetics, Globe Institute, University of Copenhagen, Copenhagen, Denmark. [8]Department of Geosciences and Natural Resource Management, University of Copenhagen, Copenhagen, Denmark. ✉e-mail: abbas@space.dtu.dk

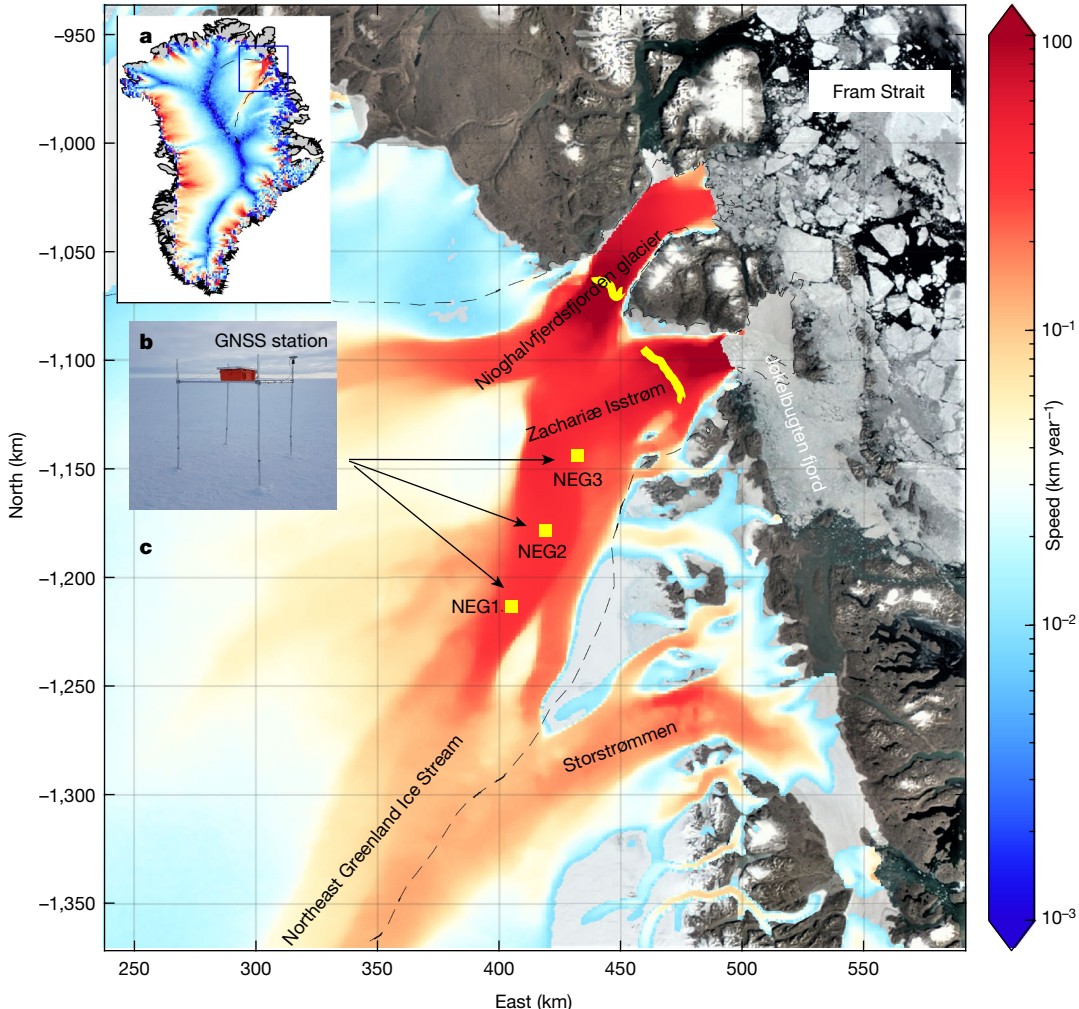

**Fig. 1 | Ice speed and location names. a**, Map of ice speed in 2007. The dashed black line denotes the combined ZI and NG drainage area. The blue box denotes the area shown in **c**. **b**, Photo of the NEG2 station setup, which consists of an antenna, receiver (in the orange box) and solar panel (on top of the box) (source and permission: Shfaqat Abbas Khan). All are situated on a platform about 2 m above the ice surface. **c**, A Landsat-8 image from 2017 is used as the background. The colour denotes the satellite-derived surface speed. The locations of GNSS stations, NEG1, NEG2 and NEG3, are marked by yellow squares. The thick yellow curve denotes the grounding line. The image was prepared using MATLAB R2021a software.

observe a flow acceleration of $4.9 \pm 0.3$ m year$^{-2}$. At NEG2, the observed acceleration is $4.5 \pm 0.3$ m year$^{-2}$ and it is $2.7 \pm 0.2$ m year$^{-2}$ at NEG1.

To map the flow acceleration over the entire NEGIS, we use the ice speed from mosaics based on ESA Sentinel-1 SAR offset tracking[29]. The ice velocity maps cover the entire NEGIS and were derived from the intensity tracking of ESA Sentinel-1 data with a 12-day repeat; the operational interferometric post processing chain was applied for the analysis[29]. Although most studies estimate the acceleration from differences between two speed mosaics, here we use a different approach. We use all available speed mosaics (provided on a grid with a spatial resolution of 500 m) to create a time series of the speed for each grid point. We then remove outliers and fit a trend to each grid point to estimate the flow acceleration (Extended Data Fig. 3 and Methods). This approach suggests that the flow acceleration propagated more than 200 km inland from 2016 to 2022 (Fig. 2d) and is fully consistent with high-precision GNSS-derived accelerations (Fig. 2a–c).

The speed-up of the NEGIS triggered by the gradual retreat of ZI's terminus (Extended Data Fig. 1 and Methods) is also responsible for dynamic thinning[4]. We use satellite and airborne altimetry data to detect changes in the ice surface elevation along the main trunk. We estimate annual elevation change rates over the ice surface from April 2011 to April 2021 using radar altimetry data from ESA's Earth

Explorer CryoSat-2 mission[31]. To better resolve elevation changes along the ice-sheet margin, we use laser altimetry observations from NASA's Operation IceBridge Airborne Topographic Mapper (ATM) flights from April 2011 to April 2019 (ref. [32]) and Ice, Cloud, and Land Elevation Satellite-2 (ICESat-2) data from October 2018 to April 2021 (refs. [33,34]). We find a thinning of more than 30 m from 2011 to 2021 near the ZI calving front (Fig. 3j). The entire lower portion of ZI (between 0 and 50 km inland) thinned between 8 and 35 m over the past decade (Fig. 3k). Between 100 and 200 km upstream along the main trunk, the NEGIS thinned between 2 and 3 m from 2011 to 2021. Thinning is traced up to 250 km inland and the pattern of thinning suggests a large amount of thinning along the main trunk of the NEGIS. The neighbouring glacier, NG, likewise shows considerable thinning of 5–15 m near the grounding line.

## Numerical ice flow model

To determine whether the observed changes are captured by numerical models, we use a high-resolution ice flow model[8] to simulate changes in the NEGIS from 2007 to 2017 (Methods). The model from Choi et al.[8] used a Budd friction law (linear viscous) and captured the observed ice speed changes and elevation changes, but only in the lower region of

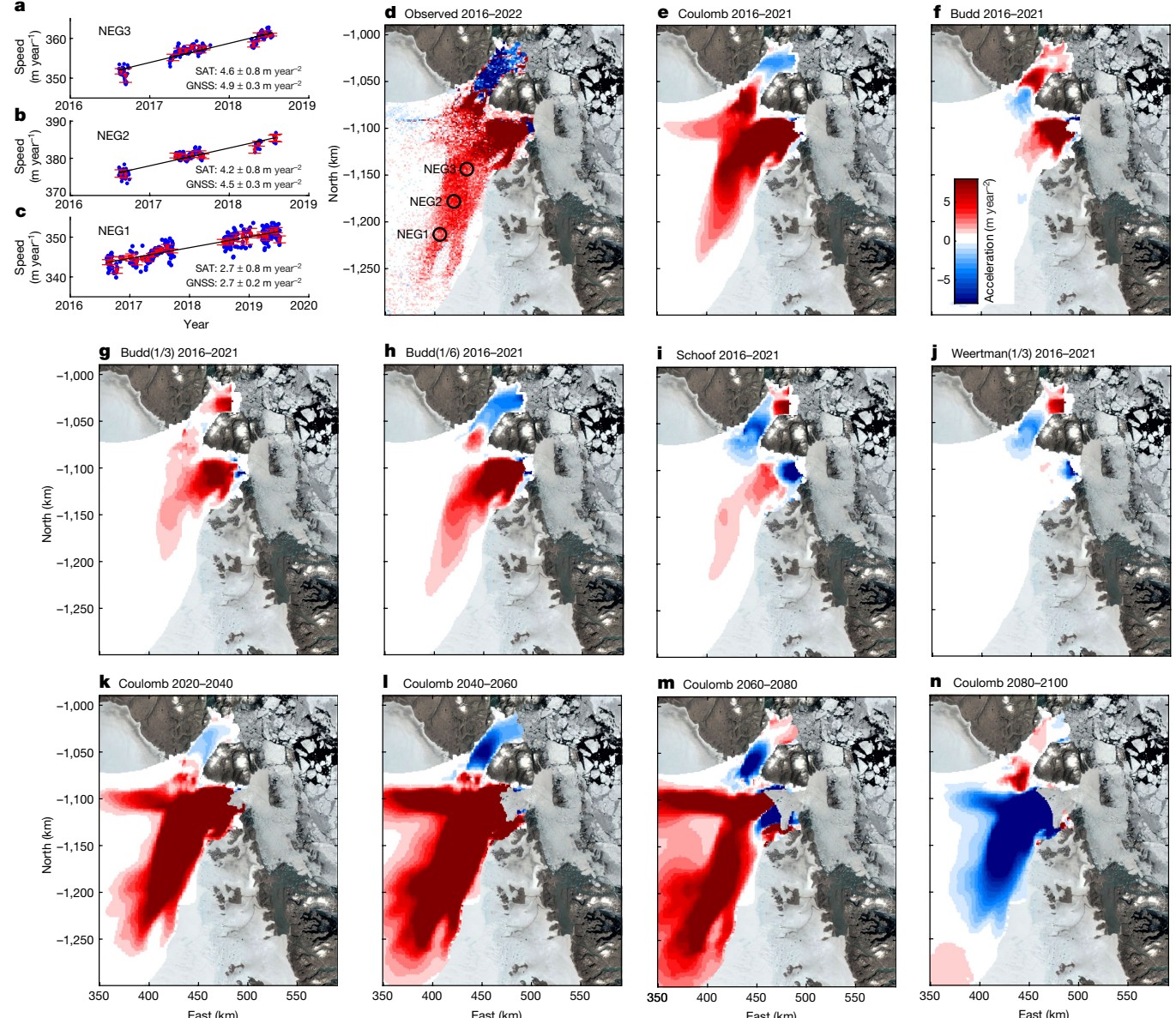

**Fig. 2 | Observed and modelled accelerations in ice speed.** Time series of daily solutions of ice speed at NEG3 (**a**), NEG2 (**b**) and NEG1 (**c**). The red error bars denote the mean monthly speed and associated root-mean-square error. The black line denotes the best-fitting trend. The mean flow accelerations in m year$^{-2}$ from GNSS data (GNSS) and from mosaics based on ESA Sentinel-1 SAR offset tracking (SAT) are listed for each site. **d**, Map of the observed ice flow acceleration from 2016 to 2022. **e**, Map of modelled ice flow acceleration in m year$^{-2}$ from 2016 to 2021 using NorESM1, RCP4.5 and regularized Coulomb friction law. **f**, Modelled ice flow using Budd friction law (linear viscous).

**g**, Modelled ice flow using Budd friction law with a friction coefficient of 1/3. **h**, Modelled ice flow using Budd friction law with a friction coefficient of 1/6. **i**, Modelled ice flow using Schoof friction law. **j**, Modelled ice flow using Weertman friction law with a friction coefficient of 1/3. Map of modelled ice flow acceleration in m year$^{-2}$ using NorESM1, RCP4.5 and regularized Coulomb friction law for the periods 2020–2040 (**k**), 2040–2060 (**l**), 2060–2080 (**m**) and 2080–2100 (**n**). Colour bar: red denotes acceleration and blue denotes slowdown. The image was prepared using MATLAB R2021a software.

the NEGIS close to the ice margin. This basal friction parameterization, however, does not capture the observed acceleration in the flow speed deep inland that was detected by satellite data (Fig. 2d,f). We tested several other friction laws (such as Budd with different velocity exponents[35], regularized Coulomb[36], Schoof[37], Weertman[38]) while forcing the ice-front retreat based on observed terminus positions. We found that the friction laws that are almost plastic are able to reproduce deep inland acceleration and thinning with remarkably good agreement (Fig. 2d,e), as well as reproducing the observed mass loss from 2011 to 2021 (Fig. 4). The plastic bed conditions are consistent with the results from previous studies[39], which may be due to bed roughness or the presence of sediments underlying outlet glaciers[40].

## Implications for future evolution of NEGIS

We then use these numerical ice flow models for forecast simulations, using forcings similar to those used in ref. [8] but with the 'nearly plastic' friction laws that achieved a substantially better fit during the hindcast period. Although the deformation of subglacial sediments[41] and the subglacial hydrology system[42] could potentially change basal conditions in the future, here we assume that the friction coefficient is constant in time for all simulations. We force the ice flow model using surface mass balance (SMB, the net balance between the processes of accumulation and ablation) anomalies from MAR (Regional Atmosphere Model)[43].

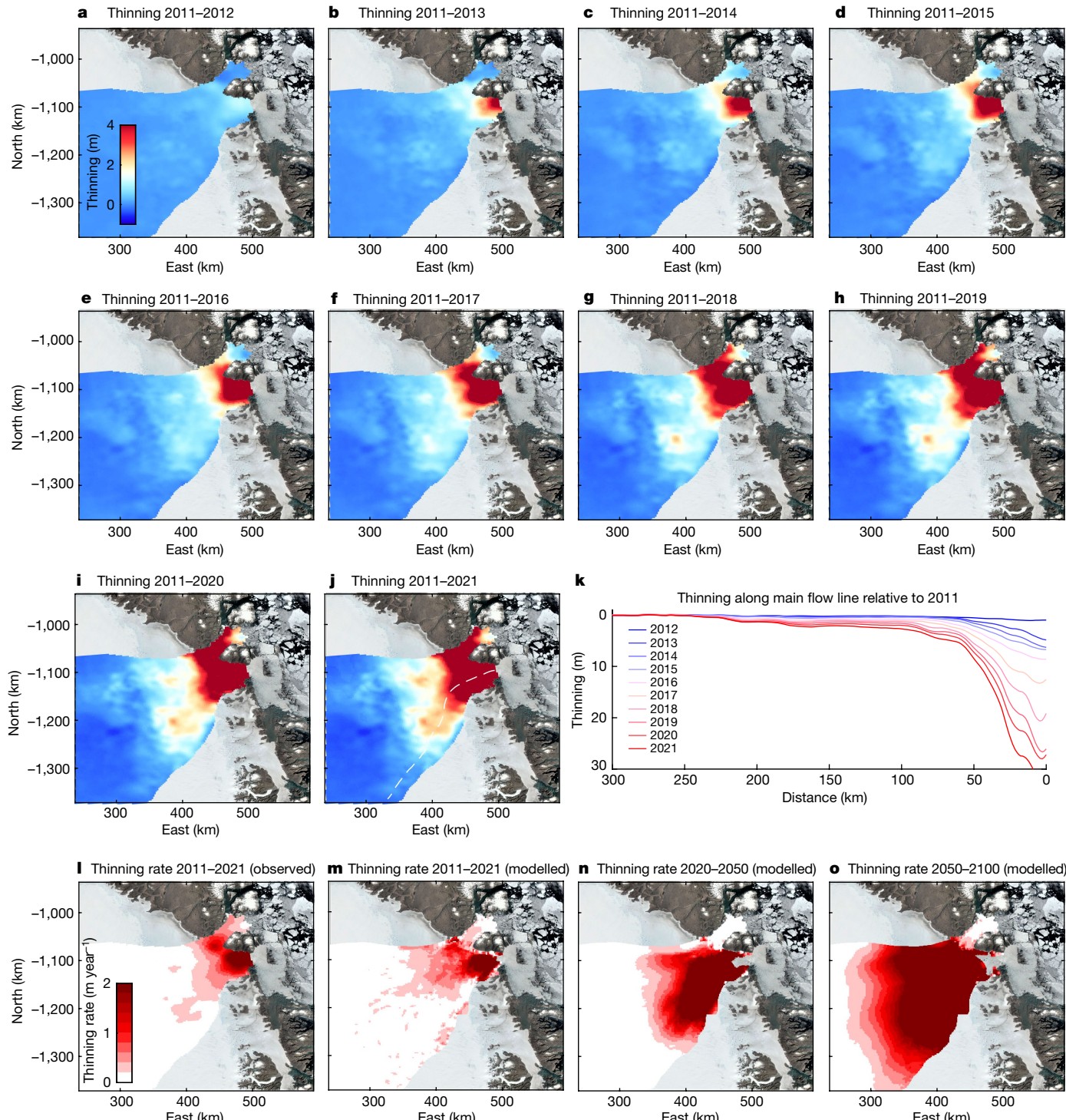

**Fig. 3 | Thinning of NEGIS based on airborne and satellite altimetry.** Thinning of the ice (April to April) during 2011–2012 (**a**), 2011–2013 (**b**), 2011–2014 (**c**), 2011–2015 (**d**), 2011–2016 (**e**), 2011–2017 (**f**), 2011–2018 (**g**), 2011–2019 (**h**), 2011–2020 (**i**) and 2011–2021 (**j**). The white line denotes the main ice flow line. **k**, Thinning along the flow line from April 2011 to April 2021. **l**, Mean observed thinning rate from 2011 to 2021. Mean modelled thinning rate during 2011–2021 (**m**), 2020–2050 (**n**) and 2050–2100 (**o**). The model used NorESM1, RCP4.5 and the regularized Coulomb friction law. The image was prepared using MATLAB R2021a software.

We model changes in the SMB and ocean forcing until 2100 using the output from three general circulation models (GCMs), MIROC5, CanESM2 and NorESM1, from the Coupled Model Intercomparison Project Phase 5 (CMIP5), based on Representative Concentration Pathway 4.5 (RCP4.5) and RCP8.5 scenarios, respectively[44,45]. The MAR simulations are carried out with a fixed ice surface elevation and, therefore, we apply a correction for the SMB anomalies using the gradient method to account for the elevation-SMB feedback.

The flow model results show an increase in thinning rates from 2021 to 2050, and they indicate that widespread thinning will migrate further into the interior of the NEGIS. From 2050 to 2100, the thinning rate continues to increase and spreads farther inland. The entire region

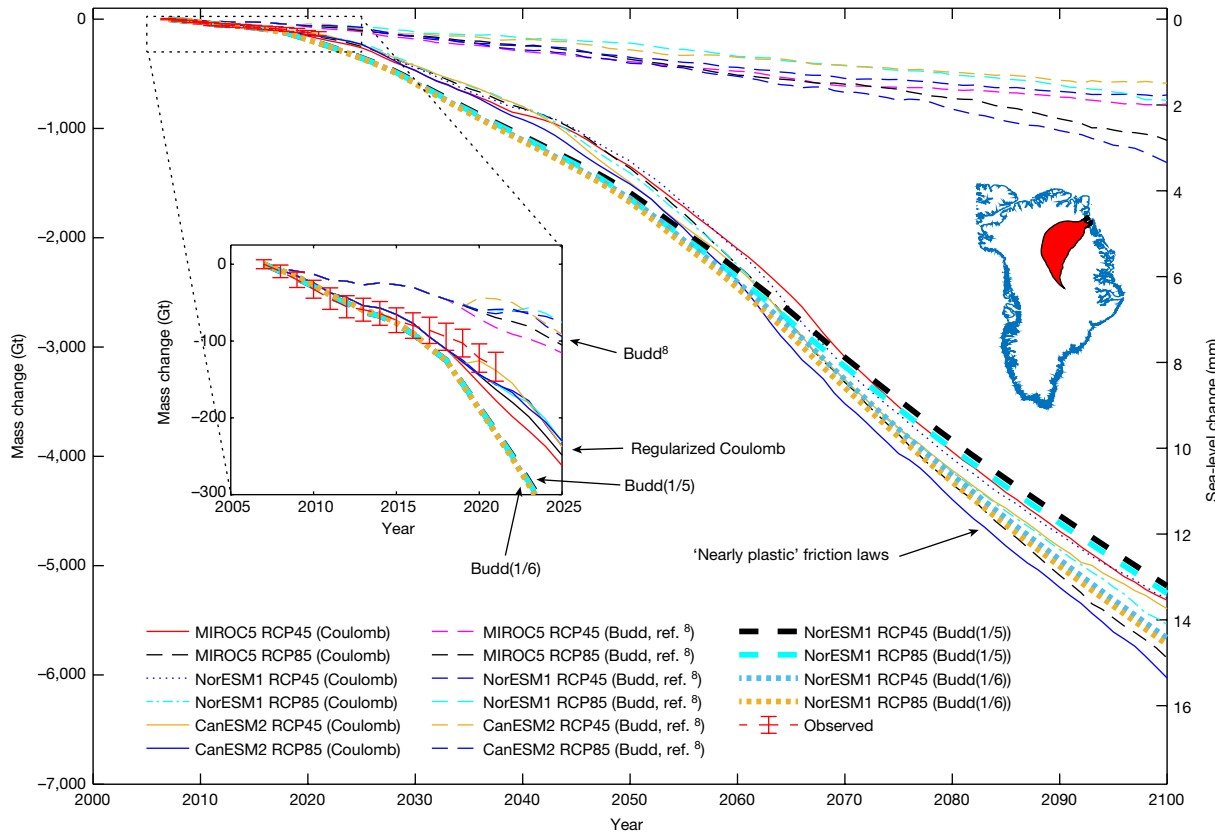

**Fig. 4 | Modelled and observed ice mass loss.** Changes in ice mass from 2007 to 2100 for 16 different models. The left vertical axis shows ice mass changes in Gt. The right vertical axis shows the ice loss converted into sea-level change in mm. Ice mass changes are estimated for the ZI and NG drainage area, which is shown in red on the map of Greenland. The observed mass change is based on airborne and satellite altimetry. The image was prepared using MATLAB R2021a software.

up to 250 km inland, an area of more than 20,000 km$^2$, experiences a thinning rate of 2–4 m year$^{-1}$ (Fig. 3o), which was only observed at the frontal portions of ZI and NG during the past decade (Fig. 3i). The model suggests a change in the flow pattern of NG and ZI (Fig. 2e,k–m). NG will change its flow direction and flow directly into the outlet of ZI (Fig. 2k–m).

The cumulative basin-wide ice mass losses for various GCMs and nearly plastic friction laws, under RCP4.5 and RCP8.5, are shown in Fig. 4. The differences between the model elevation, ice speed and mass loss for the three GCMs (MIROC5, CanESM2 and NorESM1) are relatively small. The choice of friction law, on the other hand, has a greater impact on our results than the choice of GCM. All ice flow model simulations lead to a collapse of the frontal portion of ZI, up to 30 km upstream, with notable dynamic thinning reaching the interior of the GrIS. These new insights in ice flow model simulations have important implications for the contribution of the NEGIS to sea-level rise. The projection of cumulative mass loss using the Budd friction law (linear viscous) has a sea-level equivalent of 1.5 to 3.3 mm by 2100 (ref. [8]). According to our new ice flow models, which are able to reproduce deep-inland observed accelerations and thinning, the mass loss increases by a factor of 5 to between 13.5 and 15.5 mm (Table 1).

## Conclusions

After being almost stable from the late 1970s to the early 2000s, ZI entered a phase of retreat around 2004. During the 2000s, ZI continued to retreat. A notable change occurred in 2012, when the ice shelf collapsed and the ice flow accelerated. The continuously warmer air and ocean temperatures combined with a downward-sloping marine-based bed have led to destabilization that has continued throughout the 2010s

into 2021. Our model simulations, constrained by observations over the entire length of the glaciers, suggest that widespread thinning and flow speed-up will continue throughout this century at an accelerated rate.

Our study shows that ice flow accelerations in the interior of an ice sheet (>100 km inland) obtained from in situ GNSS observations or satellite data greatly improve the predictive skills of models used to

**Table 1 | Modelled sea-level rise**

| Friction law | RCP | GCM | SLR (mm) |
|---|---|---|---|
| Budd[8] | 4.5 | MIROC5 | 2.0 |
| Budd[8] | 4.5 | CanESM2 | 1.5 |
| Budd[8] | 4.5 | NorESM1 | 1.8 |
| Budd[8] | 8.5 | MIROC5 | 2.8 |
| Budd[8] | 8.5 | CanESM2 | 3.3 |
| Budd[8] | 8.5 | NorESM1 | 1.9 |
| Regularized Coulomb | 4.5 | MIROC5 | 13.5 |
| Regularized Coulomb | 4.5 | CanESM2 | 13.7 |
| Regularized Coulomb | 4.5 | NorESM1 | 13.5 |
| Regularized Coulomb | 8.5 | MIROC5 | 14.9 |
| Regularized Coulomb | 8.5 | CanESM2 | 15.4 |
| Regularized Coulomb | 8.5 | NorESM1 | 15.5 |
| Budd with coefficient of 1/5 | 4.5 | NorESM1 | 13.2 |
| Budd with coefficient of 1/5 | 8.5 | NorESM1 | 13.4 |
| Budd with coefficient of 1/6 | 4.5 | NorESM1 | 14.4 |
| Budd with coefficient of 1/6 | 8.5 | NorESM1 | 14.6 |

Contribution to sea-level rise (SLR) in mm by 2100 for different models.

# Article

project future sea-level rise. The use of GNSS observations and satellite data to detect inland flow accelerations could be crucial for other large glacier systems, such as Pine Island Glacier or Thwaites Glacier in Antarctica[13–18], which have shown substantial changes in the flow speed and thinning near their margins in recent decades[46,47]. Furthermore, farther south in Greenland, another substantial marine basin has also started to disintegrate. Jakobshavn Isbræ (JI) drains 6% of the GrIS drainage area and contributed 4.2 ± 0.5 mm to sea-level rise from 1875 to 2012 (ref. [48]). The glacier retreated more than 40 km, with a large collapse of the floating tongue in 2000 (ref. [48]). Similar to ZI, JI is retreating along a negative bed slope (a bed that deepens inland)[49,50]. Recent model projections for JI have been calibrated using observations from the lower sectors[8,25,49]. We posit that these projections could be underestimations. Taking into account the propagation of inland thinning more completely in the simulations would increase the estimated future contributions of Greenland and Antarctica to sea-level rise and would markedly reduce uncertainty in future sea-level-rise estimates.

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

# Methods

## Calving front positions

We map the positions of the calving fronts of ZI and NG using aerial and Landsat 5–8 optical satellite imagery[51,52] from 2000 to 2021 (Extended Data Fig. 1). For ZI, the most notable changes occurred between 2011 and 2013, when a large portion of its floating extension collapsed, resulting in a retreat of 10 km (ref. [53]). This collapse has been followed by a steady retreat of about 700 m year$^{-1}$. NG lost large sections of its floating tongue from 2002 to 2004; after this, calving events of the main floating tongue have been less frequent. However, in 2020, the northern section of NG completely collapsed, releasing more than 120 km$^2$ of floating shelf ice into the ocean (Extended Data Fig. 1).

## GNSS data processing

We process the GNSS data using the GIPSY-OASIS software package with high-precision kinematic data processing methods[27] and with ambiguity resolution using the orbit and clock products of the Jet Propulsion Laboratory (JPL). We use GIPSY-OASIS version 6.4, which was developed at the JPL[30]. We use JPL final orbit products, which include satellite orbits, satellite clock parameters and Earth orientation parameters. The orbit products take the satellite antenna phase centre offsets into account. The atmospheric delay parameters are modelled using the Vienna Mapping Function 1 (VMF1) with VMF1 grid nominals[54]. Corrections are applied to remove the solid Earth tide and ocean tidal loading. The amplitudes and phases of the main ocean tidal loading terms are calculated using the Automatic Loading Provider (http://holt.oso.chalmers.se/loading/), which is applied to the FES2014b ocean tide model including correction for the centre of mass motion of the Earth owing to the ocean tides. The site coordinates are computed in the IGS14 frame[55]. We convert the Cartesian coordinates at 15-s intervals into local up, north and east coordinates for each GNSS site monitored at the NEGIS surface. An example of a 15-s solution is shown in Extended Data Fig. 2a–c.

## GNSS-derived surface speeds and their uncertainties

We use the 15-s solution to estimate daily solutions of the ice surface speed (blue dots, Extended Data Fig. 2d–f). The time series have been screened for outliers. To remove outliers, we fit and remove a trend to each time series of speed, latitude and longitude. We estimate the mean of detrended speed, latitude and longitude and define outliers as values greater than three standard deviations from the mean. For NEG1, we removed in total eight data points or (8/387 = 0.02) 2% of data. For NEG2, we removed in total four data points or (4/195 = 0.02) 2% of data and 0% for NEG3. We use daily solutions of the ice surface speed (screened for outliers) to estimate the monthly mean ice speed (red error bars). We estimate the root-mean-square of each monthly mean to assign uncertainties. The black lines denote the trends (using least square adjustment) that best fit the observed ice speeds and represent mean flow accelerations from 2016 to 2019 of 2.7 ± 0.2 m year$^{-2}$ at NEG1, 4.5 ± 0.3 m year$^{-2}$ at NEG2 and 4.9 ± 0.3 m year$^{-2}$ at NEG3 (corrected for downhill acceleration).

## Surface speed and acceleration from mosaics based on ESA Sentinel-1 SAR offset tracking

We derive ice speeds from mosaics based on ESA Sentinel-1 SAR offset tracking obtained from https://dataverse01.geus.dk/dataverse/Ice_velocity. The ice velocity maps of the NEGIS are derived from the intensity tracking of the ESA Sentinel-1 data with a 12-day repeat; the operational interferometric post-processing chain is applied for the analysis[29]. We use all available speed mosaics (provided on a grid with a spatial resolution of 500 m) and associated standard deviation of the underlying shift maps generated by the offset tracking to create a time series of speed for each grid point (Extended Data Fig. 3b–d). We then remove outliers. To remove outliers, we fit and remove a trend to

each time series of speed and estimate the mean. We define outliers as values greater than three standard deviations from the mean. For each grid point, we remove from 0 to 1.5% of data corresponding to 0 to 3 data points. Next, we use the screened time series at each grid point and fit a trend that represents the flow acceleration (the black line in Extended Data Fig. 3b,c). For each grid point, we use the least-squares fit to estimate the acceleration and associated uncertainty. Extended Data Fig. 3a shows the mean acceleration from 2016 to 2022 in m year$^{-2}$ at each grid point. The uncertainty is ± 0.7 m year$^{-2}$.

## Downhill correction for GNSS

We use mosaics based on ESA Sentinel-1 to estimate the downhill correction of the flow acceleration at GNSS stations. We estimate the time series of the ice speed at two pixels on a velocity map on a 0.5 × 0.5-km grid. Pixel 1 is located at the GNSS starting position (when the station was deployed) and pixel 2 is located at the GNSS end-point position. The difference between the two time series is used to estimate the downhill correction for the flow acceleration. The uncertainty level in Extended Data Fig. 3a is ±0.7 m year$^{-2}$. However, two pixels close to each other (such as 1,000 m apart) experience almost the same noise. Thus, when estimating the difference between two neighbouring time series, the noise is reduced to ±0.3 m year$^{-2}$. To estimate the downhill correction of the flow acceleration, we fit a trend to the time series of the differences between two pixels.

Our corrections of the flow acceleration, which compensate for the downhill movement of the GNSS stations, are as follows: 0.06 ± 0.18 m year$^{-2}$ at NEG1, 0.14 ± 0.29 m year$^{-2}$ at NEG2 and 0.12 ± 0.30 m year$^{-2}$ at NEG3.

## Elevation changes from CryoSat-2

To estimate elevation changes over the ice surface, we use a regular grid with a resolution of 500 × 500 m that covers the NEGIS. We denote the centre of each grid point as $\mathbf{C}(x_0, y_0)$. For each grid point, we select CryoSat-2 data with coordinates $\mathbf{P}(x_i, y_i)$, with a maximum distance of 1,000 m from $\mathbf{C}$. The CryoSat-2 data points with coordinates $\mathbf{P}(x_i, y_i)$ have an elevation $h_i$ measured at time $t_i$. The index $i$ denotes the $i$th data point. We use all available CryoSat-2 data measured between July 2010 and July 2021 to create surface elevation time series at each grid point $\mathbf{C}$. Previous studies have used a third-order polynomial equation to describe changes in the elevation and a third-order polynomial equation to describe the shape of the surface[2,56,57]. However, to describe surface changes over 10 years, we fit a polynomial with a degree of 7 to describe changes in the elevation and we use a third-order polynomial equation to describe the shape of the surface. In addition, we fit a seasonal term to account for the annual surface changes. For each grid point with a centre $\mathbf{C}(x_0, y_0)$, we find the nearest data point within 1,000 m and fit a seventh-order polynomial $H(t_i)_{poly}$, a third-order surface topography polynomial $H_{topo}$ and an annual term $H(t_i)_{Annual}$:

$$H(t_i) = H(t_i)_{poly} + H_{topo} + H(t_i)_{Annual}$$

Extended Data Fig. 4a shows all the grid points on the NEGIS where we have successfully estimated the time series of $H(t)$.

Extended Data Fig. 4b,c shows two examples of elevation time series $H(t)$. P1 is a point closest to the ice margin and is located at about 590 m elevation. Extended Data Fig. 4b shows CryoSat-2 elevations corrected for the topography $H_{topo}$ (black error bars) and a combination of our best-fitting seventh-order polynomial $H(t)_{poly}$ and the annual term $H(t)_{Annual}$ (red curve). The annual term at this elevation has an amplitude of 1.41 ± 0.10 m. Extended Data Fig. 4d shows CryoSat-2 elevations corrected for topography and the annual term (black error bars), and our best-fitting seventh-order polynomial (red curve) for point P1.

Extended Data Fig. 4c shows the same information as Extended Data Fig. 4b but for the point P2, which is located at about 1,078 m elevation. Here the amplitude of the annual signal is 0.16 ± 0.08 m.

We use the best-fitting seventh-order polynomial (such as the red curve in Extended Data Fig. 4d,e) for each location shown in Extended Data Fig. 4a to estimate the annual rates of elevation changes from April to April, for example, from April 2011 to April 2012, from April 2012 to April 2013 etc.

## Elevation changes from ICESat-2

We use ICESat-2 data from October 2018 to June 2021 and estimate elevation changes over the ice surface[33]. We use the ICESat-2 Algorithm Theoretical Basis Document for Land Ice Along-Track Height (ATL06) Release 004, which was retrieved from https://nsidc.org/data/atl06 (ref. [58]).

We estimate elevation changes using the same method described in the previous section. However, to describe surface changes over about 2.5 years, we fit a third-order polynomial (as opposed to the seventh-order polynomial used for CryoSat-2 data) and we also use a third-order polynomial equation to describe the shape of the surface and a seasonal term to account for the annual surface changes. Extended Data Fig. 5a shows the grid points on the NEGIS where we have successfully estimated the ICESat-2 elevation time series.

Extended Data Fig. 5 shows two examples of ICESat-2 elevation time series $H(t)$. P3 is a point close to the ice margin and is located at about 267 m elevation (Extended Data Fig. 5b). Extended Data Fig. 5b shows ICESat-2 elevations corrected for topography (black error bars) and a combination of our best-fitting third-order polynomial and the annual term (red curve). Extended Data Fig. 5d shows ICESat-2 elevations corrected for topography and the annual term (black error bars), and our best-fitting third-order polynomial (red curve) at P3.

Extended Data Fig. 5c shows the same information as Extended Data Fig. 5b but for the point P4, which is located at about 1,071 m elevation.

We use the best-fitting third-order polynomial (red curves in Extended Data Fig. 5d,e) for each location shown in Extended Data Fig. 5a to estimate the annual rates of elevation changes from April 2019 to April 2020 and from April 2020 to April 2021.

## Elevation changes from NASA's Operation IceBridge ATM flights

We estimate elevation changes using NASA's ATM surveys in Greenland from spring 2011 to spring 2019 (ref. [32]). The ATM flights are mainly concentrated along the margin of the GrIS. To estimate elevation changes, we take the height difference between overlapping points from two different campaigns, that is, we take the height differences between the 2011 survey and the 2012 survey, between the 2012 survey and the 2013 survey etc. However, it should be noted that no survey was conducted in spring 2020. Extended Data Fig. 6 shows ATM flight lines over the NEGIS.

Extended Data Fig. 6 (lower-right panel) shows an example of elevation changes at the overlapping points from 2016 to 2017. The largest elevation change is observed near the glacier margin.

## Gridded elevation changes and their uncertainties

It is important to note that we do not merge CryoSat-2, ICESat-2 and NASA's ATM surveys when we create elevation time series. Instead, we estimate the annual elevation change rates from April to April for each dataset independently and then merge the rates.

The observed annual elevation change rates from CryoSat-2, ICESat-2 and NASA's ATM surveys are used to interpolate elevation change rates onto a regular grid of $1 \times 1$ km. The interpolation is performed using the ordinary kriging method[59,60]. We use the observed annual elevation change rates to estimate an empirical semi-variogram. We fit a model variogram to the empirical semi-variogram to take the spatial correlation of elevation change rates into account. For each grid point, we estimate the elevation change rate $dh_{i,krig}$ and the associated error $\sigma_{i,krig}$.

We correct the observed ice surface elevations for bedrock movement caused by elastic uplift owing to present-day mass changes and long-term past ice changes (glacial isostatic adjustment (GIA)). We correct for GIA using the GNET-GIA empirical model[61]. For each grid point, we estimate the GIA uplift rate $dh_{GIA}$ and the associated uncertainty $\sigma_{GIA}$. We correct for the elastic uplift of the bedrock by convolving ice loss estimates (from CryoSat-2, ATM and ICESat-2) with the Green's functions derived by Wang et al.[62] for the elastic Earth model iasp91 with a refined crustal structure taken from CRUST 2.0. For each grid point, we estimate the elastic uplift rate $dh_{elas}$ and the associated uncertainty $\sigma_{elas}$. The elevation change rate for each grid point is

$$dh_i = dh_{i,krig} - dh_{i,elas} - dh_{i,GIA}$$

and the associated uncertainty is

$$\sigma_i = \sqrt{\sigma_{i,krig}^2 + \sigma_{i,elas}^2 + \sigma_{i,GIA}^2}$$

Extended Data Fig. 7 shows annual elevation change rates from April to April for the period from April 2011 to April 2021 that are corrected for GIA and elastic uplift. Extended Data Fig. 7 (lower panels) shows the uncertainties associated with the annual elevation change rates from April to April.

## Ice flow modelling

We use the Ice-sheet and Sea-level System Model (ISSM)[63], a finite-element ice flow model, to model the NEGIS. The horizontal mesh resolution varies from 200 m near the ice front to 20 km inland, and it is vertically extruded into four layers. The model is based on a 3D higher-order approximation of the full Stokes model to include vertical shear for the stress balance; this is a good approximation for both fast-moving and slow-moving regions[64,65]. We use the surface and bed geometry from BedMachine Greenland[50] (version 3). We infer the ice viscosity parameter over floating ice and basal conditions under grounded ice using inversions. To infer the initial basal conditions, we use the Budd friction law and invert for the friction coefficient using surface velocities from 2007 to 2008 (ref. [66]). We then analytically calculate the friction coefficient for a regularized Coulomb friction law that produces the same basal stress. The friction coefficient is kept constant in time during the simulations. Previous studies have shown that the choice of friction law can have a considerable impact on simulations[67–70]. However, here we select the friction law based on observations.

The model uses the level-set-based moving boundaries to track ice front positions. We use the von Mises tensile stress calving law to calculate the calving rate[71,72]. We calibrate the stress threshold of the calving law to match the observed ice front retreat from 2007 to 2017. We use the same stress threshold values, 1 MPa for grounded ice and 150 kPa for floating ice, used by Choi et al.[24]. The calibrated stress thresholds for two glaciers (NG and ZI) are assumed to be constant for all simulations.

Using these nearly plastic friction laws, and the same atmospheric and oceanic forcing as Choi et al.[24], we calibrate the calving law to qualitatively match the observed front changes from 2007 to 2017 (ref. [72]). For hindcast simulation, the model is forced by monthly SMB data from the Regional Atmospheric Climate Model (RACMO) version 2.3 (ref. [73]). We apply ocean forcing that considers melting under floating ice for ZI and NG[74], along with the undercutting at the ice front of the terminus of ZI once it becomes grounded[75,76]. The details of the ocean forcing parameterizations can be found in Choi et al.[8]. We keep the ice temperature constant, as it is not affected by the surface temperature on the timescales considered in this study[77]. We find that the model with a regularized Coulomb law yields much better agreement with observed accelerations (Fig. 2d,e) and the observed mass loss from 2011 to 2021 (Fig. 4).

## Friction law selection

Extended Data Fig. 8 shows maps of the observed and modelled ice flow acceleration from 2016 to 2022. The observed accelerations are identical to those shown in Extended Data Fig. 3a. Here we show the

Budd friction law[35] with different exponents, the regularized Coulomb friction law[36,69], the Schoof friction law[37] and the Weertman friction law[38] with different exponents. We tested a wide range of models with different exponents; however, our model results indicate that only the regularized Coulomb friction law or the Budd friction law with exponents of 1/5 or 1/6 can produce the deep inland acceleration observed in the satellite data. For all models in Extended Data Fig. 8, we used NorESM1 and RCP4.5. The choice of the GCM and RCP has a much smaller impact than the choice of the friction law.

In Extended Data Fig. 9, we consider models that are able to reproduce deep inland acceleration. Therefore, the choice of friction laws is reduced to the regularized Coulomb friction law and the Budd friction law with exponents of 1/5 and 1/6, as all the other models were unable to generate deep inland acceleration.

Extended Data Fig. 9 shows the modelled cumulative changes in the ice mass from 2007 to 2100 for the regularized Coulomb friction law and the Budd friction law with exponents of 1/5 and 1/6. For each friction law, we model the mass change using RCP4.5 and RCP8.5, respectively. For all models, NorESM1 was used. For comparison, we include results from ref. [8] that were calculated using the Budd friction law (linear viscous). Extended Data Fig. 9 suggests that only the mass change from the regularized Coulomb friction law is within the uncertainty level of the observed mass change from 2007 to 2021. The Budd friction law with exponents of 1/5 or 1/6 slightly overestimates the mass change from 2007 to 2021. However, we note that, by 2100, the mass loss from the Budd friction law with exponents of 1/5 or 1/6 is almost the same as that from the regularized Coulomb friction law. The retreat of the terminus from 2007 to 2100 using NorESM1, RCP4.5 and regularized Coulomb friction law is shown in Supplementary Video 1.

## Data availability

CryoSat-2 data are available at https://earth.esa.int/eogateway/catalog/cryosat-products. NASA's Operation IceBridge ATM data from April 2011 to April 2019 are available at https://nsidc.org/data/ilatm2. The ICESat-2 data are available at https://nsidc.org/data/icesat-2. Ice surface velocity and BedMachine Greenland are freely available from the National Snow and Ice Data Center (NSIDC). RACMO SMB and runoff information can be accessed at https://zenodo.org/record/3367211#.YyrIObTP23B, and MAR SMB and runoff information is available at https://mar.cnrs.fr. Ocean thermal forcing data are available at https://doi.org/10.7280/D1667W. The ice speed obtained from mosaics based on ESA Sentinel-1 SAR offset tracking is available at https://dataverse01.geus.dk/dataverse/Ice_velocity. GNSS daily solutions screened for outliers are available at https://datadryad.org/stash/share/VxgLxKo7i4_u8Fy8p4CGj37PxASIfC9hiAGpDgiHzQM. Source data are provided with this paper.

## Code availability

The ISSM is open source and is available at http://issm.jpl.nasa.gov (version 4.17, released 1 April 2020).

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

**Acknowledgements** We thank B. O. Petersen from the Joint Arctic Command (Denmark) for coordination of the logistical support, which allowed us to carry out fieldwork in northeast Greenland. S.A.K. acknowledges support from the Danish Council for Independent Research (grant no. 1026-00085B), Villum Experiment (grant no. 40718), the Danish Ministry of Climate, Energy and Utilities (project no. 2019-4542) and Carlsbergfondet (grant no. CF14-0145). Y.C. is supported by the JPL Strategic Research and Technology Development Program.

**Author contributions** S.A.K. conceived the study, analysed most of the data, especially the satellite and airborne altimetry data and GNSS data, prepared most of the figures and wrote parts of the paper. Y.C. and M.M. wrote parts of the paper and designed the ice flow model experiments. E.R. provided the data related to ocean melt rates. V.H. processed the CryoSat-2 data. J.M. analysed the surface flow speed. All authors participated in the writing of the manuscript.

**Competing interests** The authors declare no competing interests.

**Additional information**
**Correspondence and requests for materials** should be addressed to Shfaqat A. Khan.

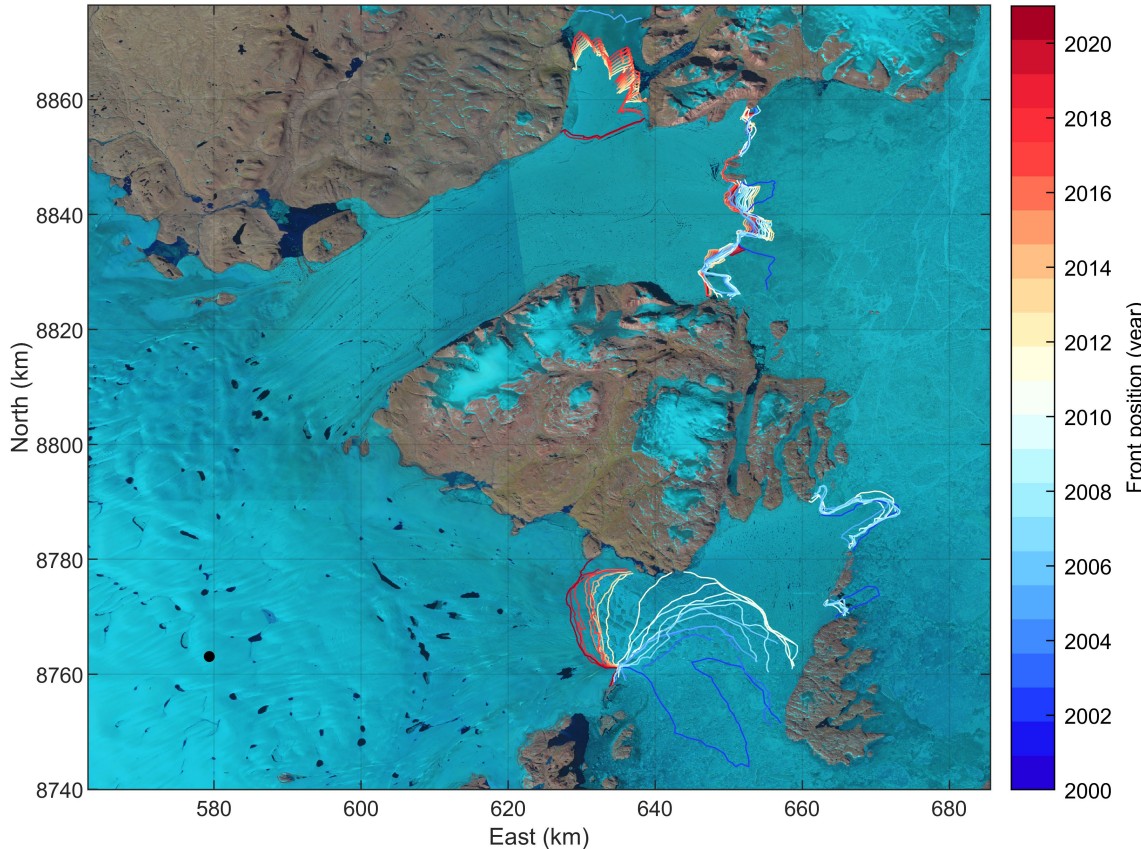

**Extended Data Fig. 1 | Terminus positions.** Frontal position from 2000 to 2021 based on Landsat 5–8 optical satellite imagery. The image was prepared using MATLAB R2021a software.

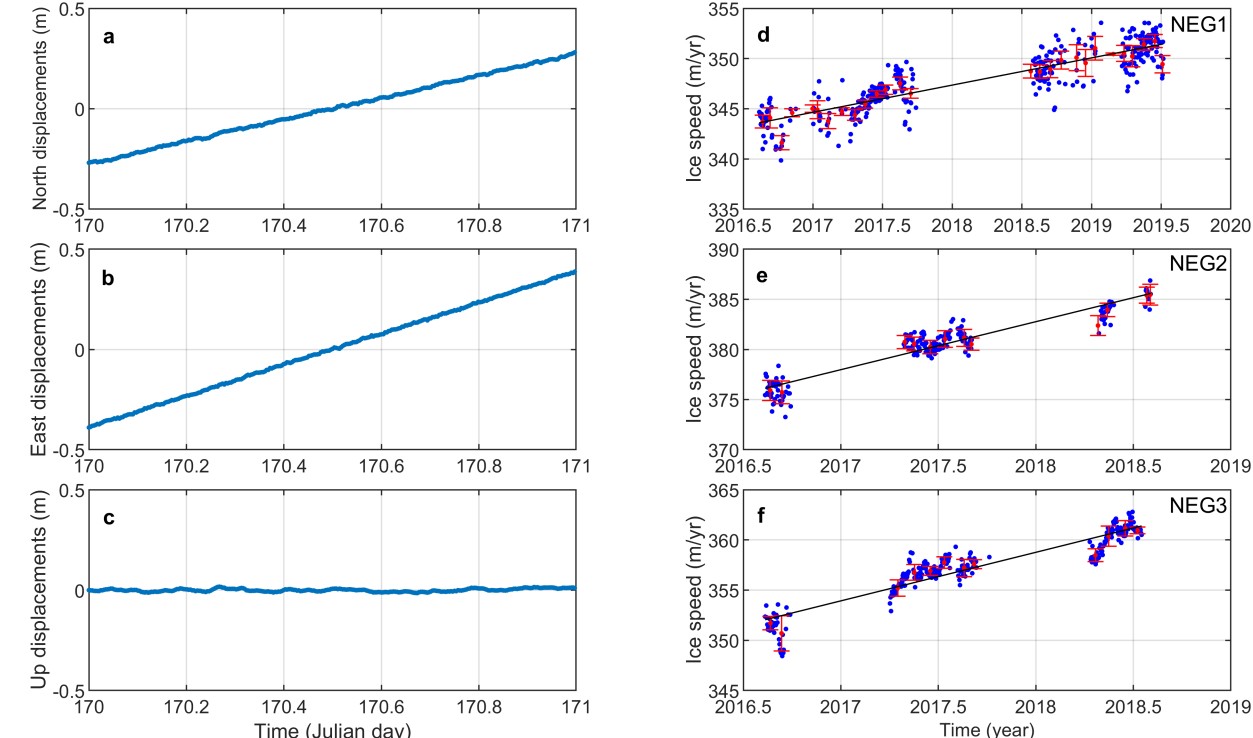

**Extended Data Fig. 2 | GNSS time series.** North (**a**), east (**b**) and up (**c**) displacement at NEG1 on 20 June 2017. Time series of daily solutions of the ice speed at NEG1 (**d**), NEG2 (**e**) and NEG3 (**f**). The red error bars denote the mean monthly ice speed. The black lines represent the best-fitting trend.

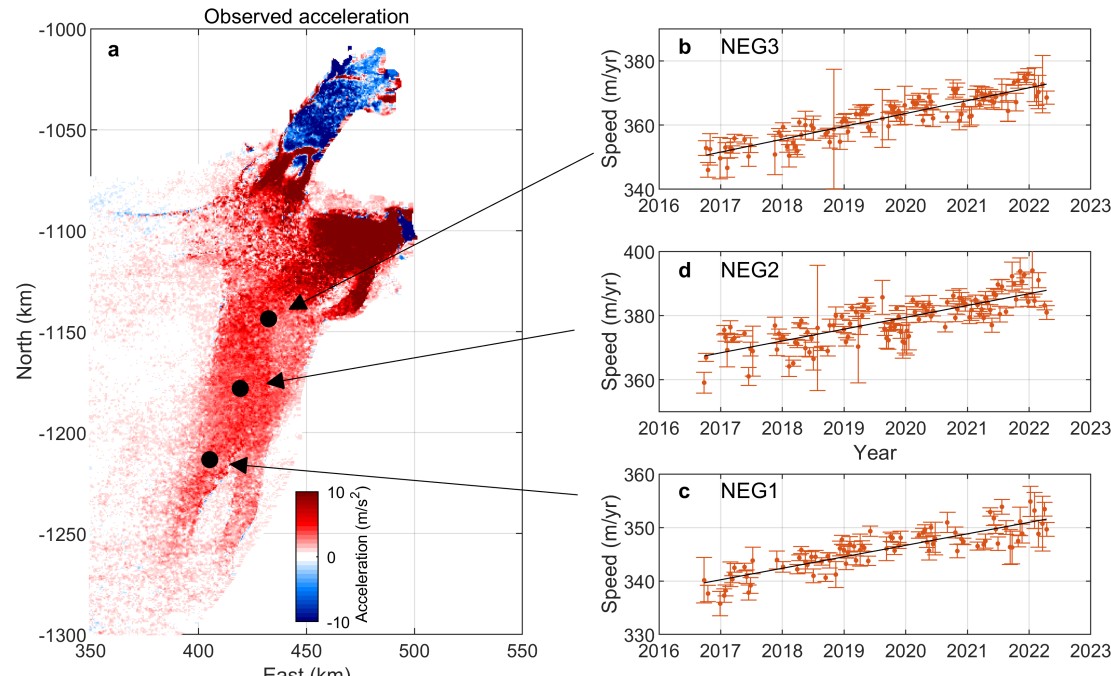

**Extended Data Fig. 3 | Time series of ice speed from Sentinel-1 data. a**, Map of the observed ice flow acceleration from 2016 to 2022. Time series of the ice speed at NEG3 (**b**), NEG2 (**c**) and NEG1 (**d**) based on the intensity tracking of ESA Sentinel-1 data with a 12-day repeat. Error bars denote mean speed and standard deviation of the underlying shift maps generated by the offset tracking. The solid black lines denote the best-fitting trend, which represent the mean flow acceleration in m year$^{-2}$.

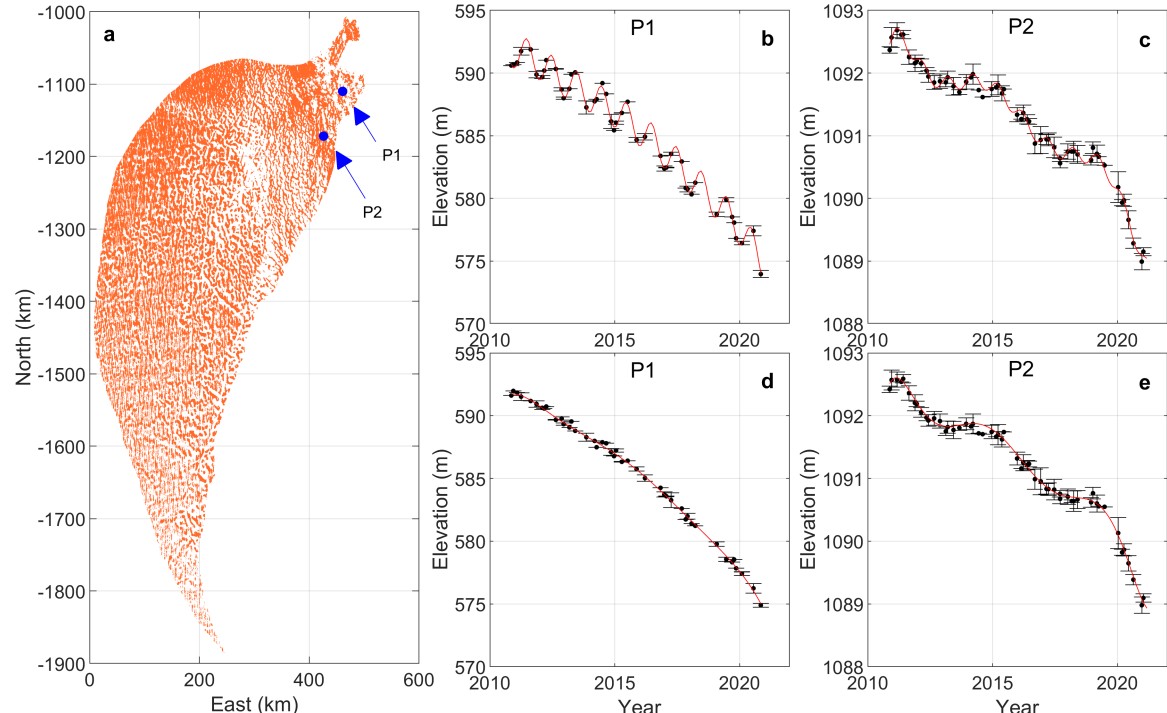

**Extended Data Fig. 4 | CryoSat-2 data coverage and elevation time series.**
**a**, Points on the NEGIS (black dots) where time series of the elevation $H(t)$ have been estimated using CryoSat-2 data. **b**, CryoSat-2 elevations at P1 corrected for topography (black error bars) and a combination of our best-fitting seventh-order polynomial and the annual term (red curve). **c**, The same as **b** but at P2. **d**, CryoSat-2 elevations at P1 corrected for topography and annual signal (black error bars) and the best-fitting seventh-order polynomial at P1 (red curve). **e**, The same as **d** but at P2.

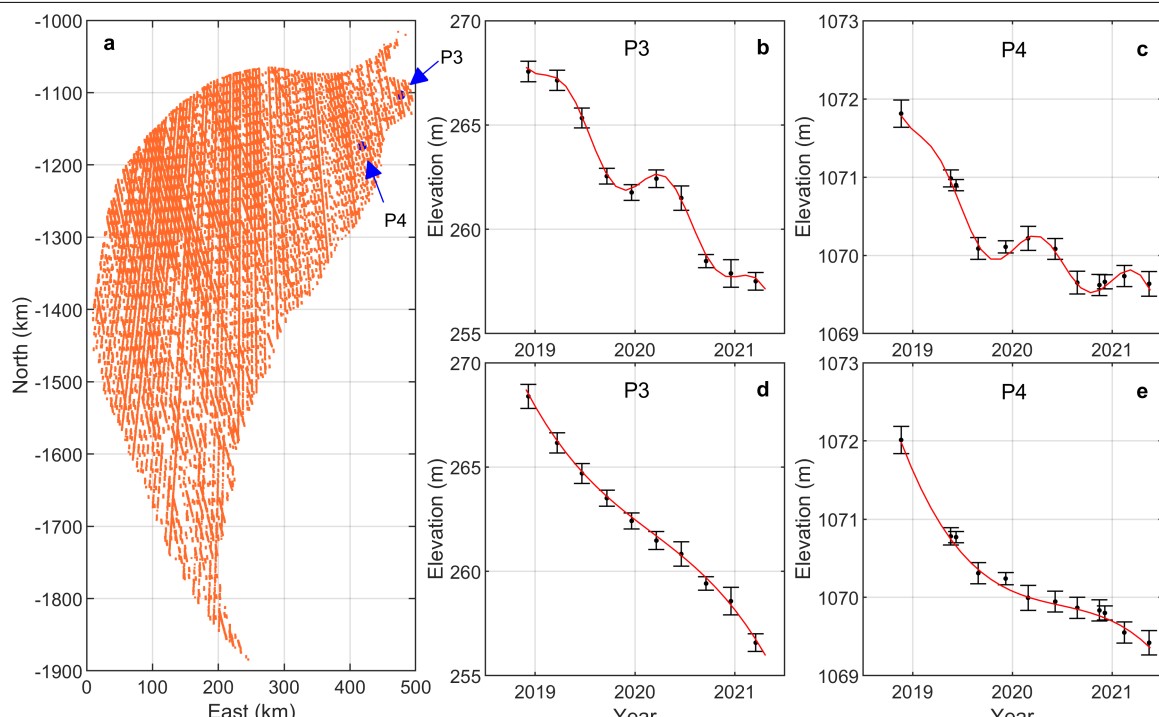

**Extended Data Fig. 5 | ICESat-2 data coverage and elevation time series.** **a**, Points on the NEGIS (black dots) where time series of the elevation $H(t)$ have been estimated using ICESat-2 data. **b**, ICESat-2 elevations at P3 corrected for topography (black error bars) and a combination of our best-fitting third-order polynomial and the annual term (red curve). **c**, The same as **b** but at P4. **d**, ICESat-2 elevations at P3 corrected for topography and the annual signal (black error bars) and the best-fitting third-order polynomial at P3 (red curve). **e**, The same as **d** but at P4.

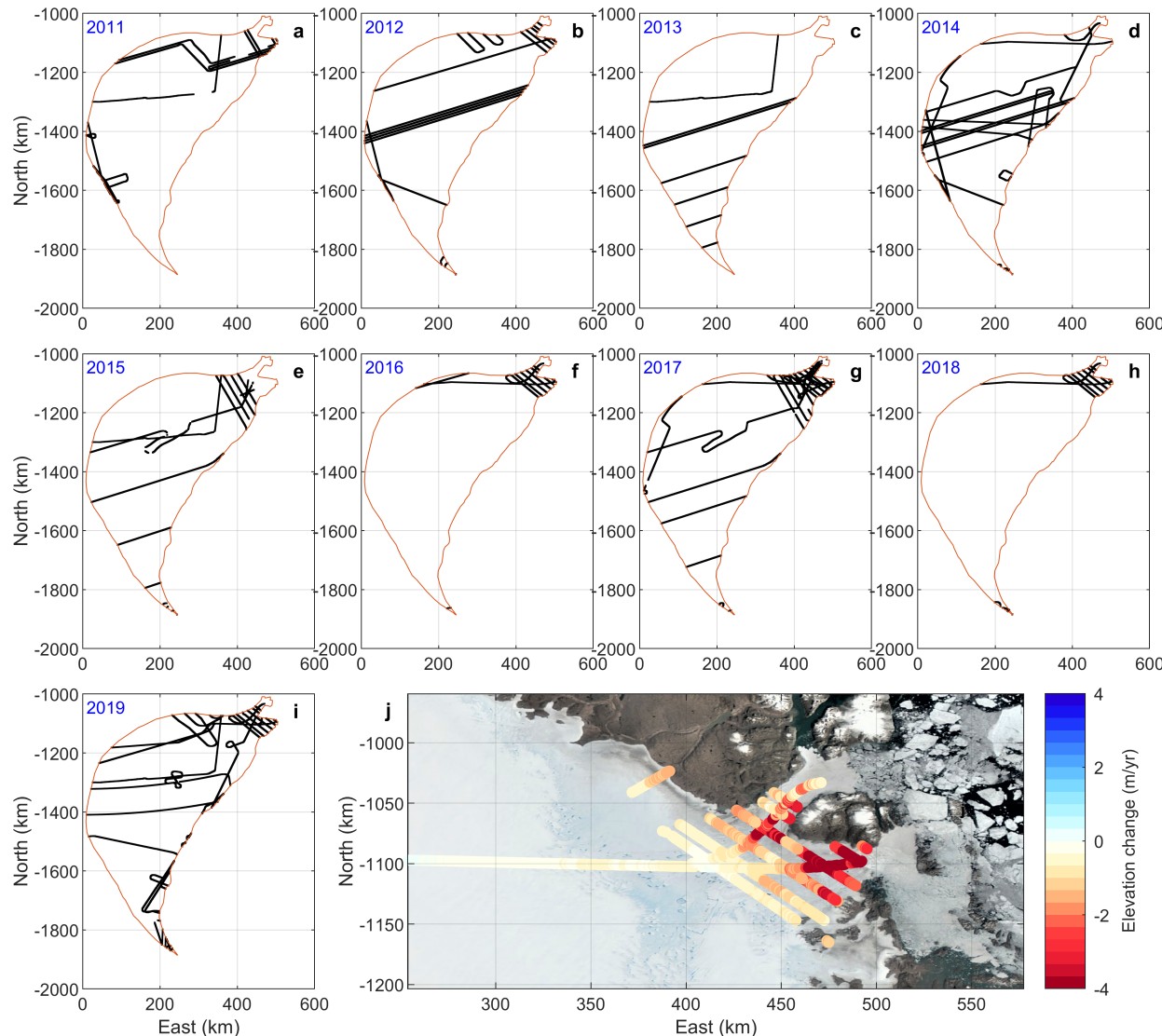

**Extended Data Fig. 6 | NASA's ATM flight lines.** NASA's ATM surveys in northeast Greenland during spring 2011 (**a**), spring 2012 (**b**), spring 2013 (**c**), spring 2014 (**d**), spring 2015 (**e**), spring 2016 (**f**), spring 2017 (**g**), spring 2018 (**h**) and spring 2019 (**i**). Black lines denote flight lines. Elevation changes in m year[−1] from NASA's ATM surveys in 2016 and 2017 (**j**). Negative values indicate surface lowering. The image was prepared using MATLAB R2021a software.

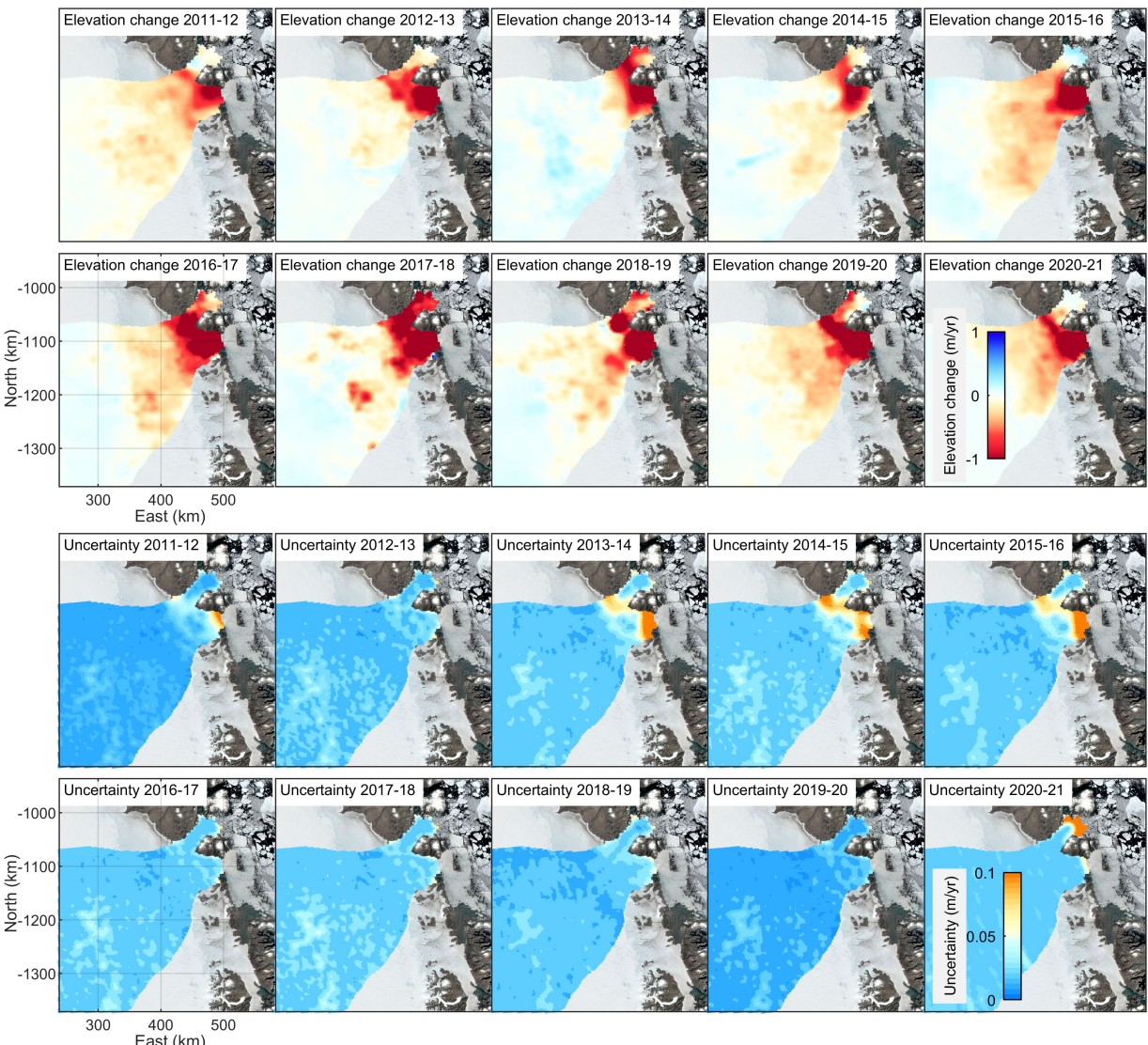

**Extended Data Fig. 7 | Elevation change rates.** Annual elevation change rates and associated uncertainties (see Methods) from 2011 to 2021. The image was prepared using MATLAB R2021a software.

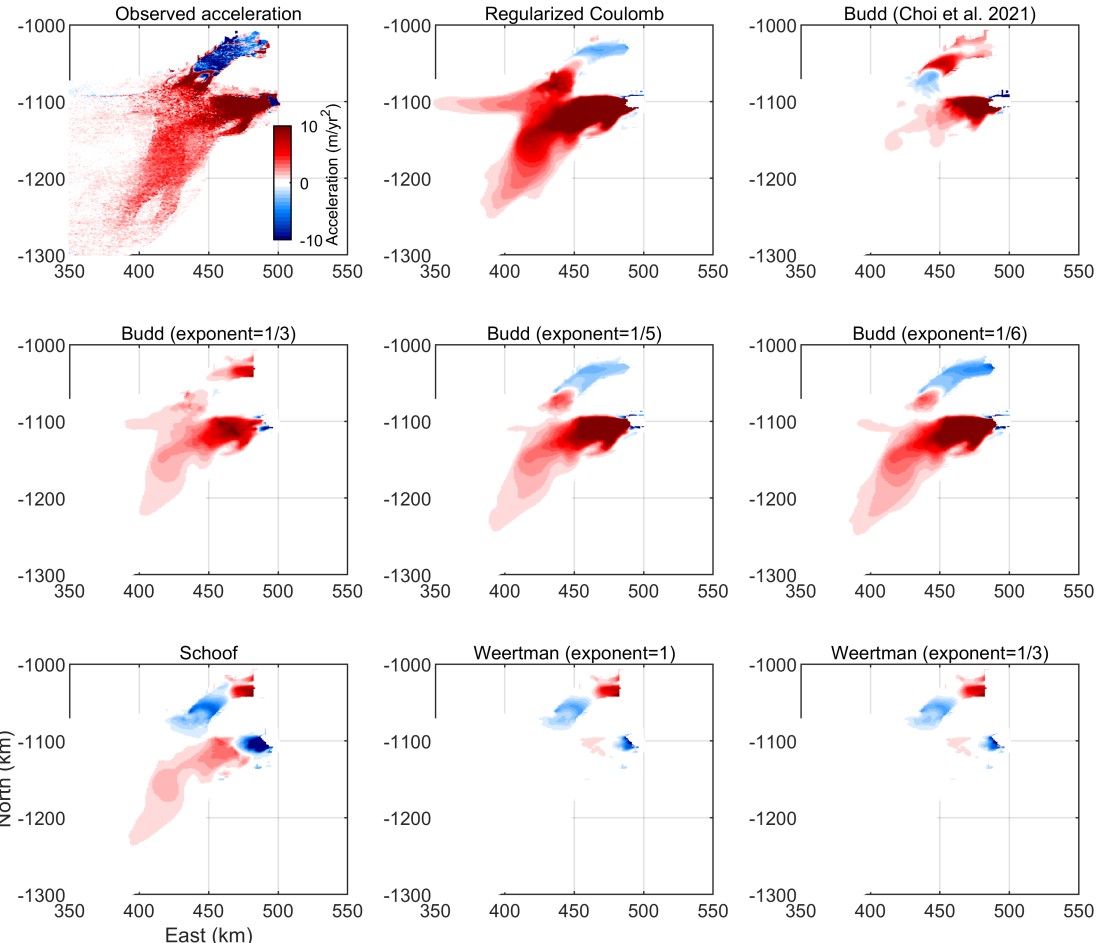

**Extended Data Fig. 8 | Observed and modelled flow acceleration.** Map of observed and modelled ice flow acceleration in m year$^{-2}$ from 2016 to 2022 using the regularized Coulomb friction law, Budd friction law as used in Choi et al.[8], Budd friction law with a coefficient of 1/3, Budd friction law with a coefficient of 1/5, Budd friction law with a coefficient of 1/6, Schoof friction law, Weertman friction law with a coefficient of 1 and Weertman friction law with a coefficient of 1/3. All models use the GCM NorESM1 and RCP4.5.

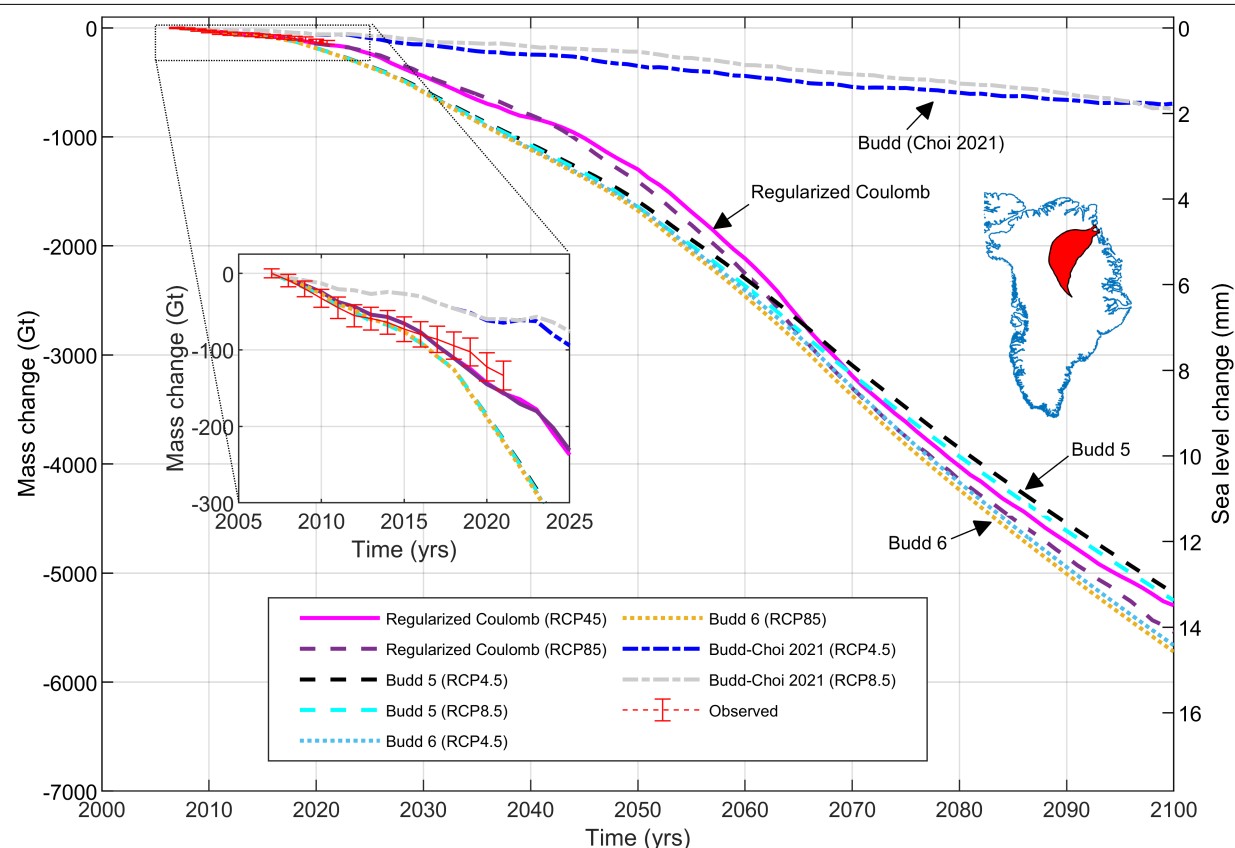

**Extended Data Fig. 9 | Time series of ice mass loss.** Observed and modelled changes in the ice mass from 2007 to 2100 for eight different models. The left *y*-axis shows ice volume changes in Gt. The right *y*-axis shows the ice loss converted into the sea-level change in mm. Cumulative ice changes are estimated for the ZI and NG drainage area, which is shown in red on the map of Greenland. The image was prepared using MATLAB R2021a software.