## [Peer Review File · Nature]

Manuscript Title: Extensive inland thinning and speed-up of North-East Greenland Ice Stream

Reviewer Comments & Author Rebuttals

Reviewer Reports on the Initial Version:

Referees' comments:

Referee #1 (Remarks to the Author):

A Summary of the key results

The manuscript conclusively shows that one of the major outlet glaciers of the Greenland Ice Sheet (GrIS) underwent very large dynamic and geometric changes in the last decade. These alarming changes will likely continue into the future, as predictions from an ice flow model show, that was calibrated with the presented measurements. This large and very remote sector of the GrIS will contribute considerably to global sea level rise.

B Originality and significance: if not novel, please include reference

This study is a milestone in ice sheet research. It is a very important and comprehensive investigation on one of the crucial topics of glaciology, climate change research, and therefore for all of humanity. Rapid changes in major Greenland Ice Sheet drainage basins have world-wide consequences by iceberg production, altering ocean fresh-water fluxes, and obviously sea level rise.

The manuscript reports on a unique data set from field measurements from one of the most challenging places on the planet. In-situ ice flow velocities are a key quantity to understand rapid ice flow which dominates the short-term changes of polar ice sheets. The acceleration and rapid thinning of the huge NEGIS ice stream is unprecedented and of highest importance for the present and future evolution and stability of the ice sheet.

C Data & methodology: validity of approach, quality of data, quality of presentation

The data set is unique and hard-earned. Obtaining good in-situ data from the Greenland ice sheet is very difficult, and from an extremely remote area like North-East Greenland even more. A consequence of this are the large data gaps in winter. Nevertheless, the data analysis has been carefully done, is convincing, and provides new and unique results. These findings are corroborated by analysis of state-of-the-art satellite data products. The model results were obtained with one of the leading ice sheet models, and properly parametrized and validated to emulate past behavior. This is crucial for predicting the future evolution of this ice stream.

D Appropriate use of statistics and treatment of uncertainties

The quantification of uncertainties is reasonable, and the discussion of measurement errors is good and sufficient.

E Conclusions: robustness, validity, reliability

The conclusions are very relevant, and undeniably robust. Using very different data acquisition systems to independently arrive at a set of consistent conclusions leaves little room for doubt of the robustness and validity of the conclusions. Obviously, the future evolution hinges on the accuracy of prescribed climate scenarios. The variability of the results by using different forcings were appropriately explored and quantified.

F Suggested improvements: experiments, data for possible revision

The manuscript should be streamlined by a very careful native speaker. Especially the last sections and the "Methods" need a lot more attention to precise and concise language.

More careful descriptions of the "models" in the main text, and especially in the "Methods", have to be given. It seems that the term "these models" (lines 104, 113, 128) refers not to different modeling codes (which I would call "models"), but merely to different subroutines/parametrizations for the calculation of basal friction within the same code base. It is crucial to very clearly state this.

In line 107 it should be stated what MAR does, rather what the acronym stands for (not all readers know French). Line 113 should start "The model results show..."

I think there is an error on line 115: only 200 km² are affected? This cannot be true, it has to be 2-3 orders of magnitude more, otherwise there would be no noticeable effect on SLR. Also the maps in Fig. 3 show an area of about 200 x 100 km that is affected by thinning.

Generally, the Figure style should be consistent, e.g. quantities plotted should be annotated by e.g. "Sea level change" (capitalized). At the moment this is random and inconsistent. Even if English has no reasonable rules for this, it should be consistent.

Figure captions should start capitalized (ATM this is not consistent).

Figure 1 is blurry. It has terrible colors for speed. A better colormap (like used in many other studies, e.g. Rignot), and a colorbar are necessary. Are the speeds shown on a linear or logarithmic color scale (for Greenland this is likely log)? Annotations NEG1-3 are barely readable, use better colors.

Figure 2 is blurry. Use consistent label capitalization ("Ice speed"). Blue circles marking positions of NEG1-3 are barely visible on red, use better colors.

Figure 3: 3k: capitalize "Cumulative". Captions starts with lower-case text. Why is this called "Cumulative thinning"? It is just thickness change. Easier to understand and more precise, too.

Showing these rates in Figure 3, is it assumed that the rate of thinning is constant? Why not show thinning from year to year, but always with the 2011 baseline?

The crucial Figure 4 is blurry. Why is this called "Cumulative mass loss"? It is just mass change/loss. Easier to understand and more precise, too.

I was quite surprised that not more care has been given to a manuscript submitted to a scientific journal. While the main text is well-written, the last section (dwellings on Jakobshavn Isbrae) seems to be a last-second addition without any streamlining or peer-review by co-authors. The figures look like a first draft, blurry, inconsistent, annotations that are difficult to read, bad color maps.

The supplementary material is full of typos (sometimes even funny), unclear formulations, and missing explanations on many aspects of the modeling exercise. All of this should be very carefully revised and clarified.

Line 92: Approximation of what, why higher order? (I understand it, but not everybody is ice sheet modeler). Better leave this away, or give a precise description. As is this is only confusing. Rather say FE, FD, FV, and say which equations are solved (likely a nonlinear diffusion equation?)

Line 59: What is the width of the stream? How representative are GPS velocities?

Line 77: Explain what you mean by this. Mass continuity requires that extension => thinning.

Line 82: "Land" capitalize like the rest

Line 86: Is this only dynamic thinning? What is the contribution of surface melt?

Line 91: the word "model" in "linear model" means something completely different from later uses. So better call this "linear fit/approximation/etc"

Line 113: "The model results show..."

Line 131: from the 1970s to ...

Line 138: wrong plural "an ice sheets "

Line 151: "sea level rise estimates."

Line 181: "de-acceleration" could be "slowdown"

G References: appropriate credit to previous work?

Previous work has been appropriately given credit. But the bibliography itself is an assembly of random styles and typesetting, DOIs are mostly missing, several typos in journal names and titles. Again, surprising that this was never peer-reviewed by the co-authors.

H Clarity and context: lucidity of abstract/summary, appropriateness of abstract, introduction and conclusions

Abstract and conclusion are clear, appropriate and well-written. The introduction is easy to understand and well crafted. The conclusions could be somewhat sharpened and streamlined.

Referee #2 (Remarks to the Author):

Review of:

Extensive inland thinning and speed-up of the Northeast Greenland Ice Stream

By Khan et al. for Nature (Manuscript ID: 2022-05-08091)

General Comment

Even though we are living in the era of satellites, ground observation provides unique data. However, both ground and satellite observations cannot look into the future evolution of glaciers, while simulations can do. Khan and coauthors combine the GPS observation, satellite observation and simulations to investigate the past and future changes of the Northeast Greenland Ice Stream, where the largest basin of Greenland locates. The data are valuable and the methodology is feasible. This work is very interesting not only to glaciologists but also to the public due to its potential contribution to global sea-level rise. I find the idea, goal and outcome of the study – the first acquisition of a set of in-situ ice flow velocity measurements and subsequent simulations on ice stream dynamics in time and space – in principal worth publishing. However, I do think that the authors could do a better job in explaining the value of their research.

Therefore, I would highly recommend a revision of the manuscript taking the following into consideration.

Firstly, the manuscript has presented the acceleration of surface ice flow speed, however the speed itself is not presented along the simulations from 2020 to 2100, while the ice flow speed is crucial to the ice flux computation, which is related to the mass balance of glaciers.

Secondly, the in-situ GPS observations are valuable. What's the type of GPS receiver/antenna for the 3 GPS stations? It's a pity that the GPS observations were not continuously from 2016 to 2020, why there are so many interruptions? How to get the observed accelerations in the missing periods? It seems that a precise point positioning method was used to analyzed the GPS data using GIPSY-OASIS, is it possible to get more precise positioning results using differential GPS method? Why using monthly averaged velocities rather than daily or weekly ones? Furthermore, there is no indication for the GPS data in "Data availability"?

Thirdly, the propagation of inland thinning would, as the authors described, increase the future contribution of Greenland to sea-level rise and reduce the uncertainty of simulation. But, how the propagation of inland thinning itself will change the basal conditions of the ice stream in the future? If the inland thinning did change the basal conditions, would it increase the uncertainty of modelling? Glacier flow speed-up is common around the Arctic (<https://doi.org/10.1007/s13131-021-1718-1>), then is it possible to quantitatively analyze the factors behind recent ice flow speed-

up of the Northeast Greenland Ice Stream?

There are also many issues that should be fixed throughout the manuscript. More details of comments are as follows.

Specific Comments

*Page 2 Line 35:

"GIS" is widely used for "Geographical Information System". I have a suggestion that "Greenland Ice Sheet (GIS)" be tuned into "Greenland Ice Sheet (GrIS)". And this change will impact the whole manuscript.

*Page 2 Line 46:

"retreat of the ice front" is not an certain reason of "rapidly speeding up", and the "rapidly speeding up" is not the inevitable result of glacier "retreat of the ice front". This sentence should be revised.

*Page 3 Line 108:

I suggest that "GCM" should be explained with a full name or references.

*Page 3 Line 114:

I think "GIS" here should be "NEGIS", because the results shown in the paragraph are limited in the NEGIS.

*Page 4 Line 117:

It is Fig. 3I that displays the observed thinning rate during the last decade, not Fig. 3m.

*Page 4 Line 138:

Repeated: "in the in the".

*Page 4 Line 151:

"reduce uncertainty in of future sea level rise" should be checked.

*Page 8 Fig.4:

All the "RPC" in the legend should be replaced with "RCP".

Why the Figure 4 only indicates error bars for the Observed curve, no error for the simulation results?

Appendix:

*Page 14 Line 379:

"august" should be "August".

*Page 18 Line 456:

In the caption of "Extended Data Fig. 7", the "(c) CryoSat-2 elevations at P2" should be "(c) CryoSat-2 elevations at P1"

*Page 22 Line 534:

The "of" in "GNET-GIA empirical model of" is needless.

*Page 27 Line 620:

In Extended Data Table 1, "RPC" should be "RCP".

Referee #3 (Remarks to the Author):

Summary of the key results:

The authors present GPS records from the northeast Greenland ice stream, which show acceleration of 2.7 - 4.9 m/yr per year 90 - 190 km inland from the coast over a three year period. The GPS observations are corroborated by satellite observations, which include both speed and height changes in the northeast Greenland ice stream's catchment. The authors attribute dynamic thinning to ice shelf disintegration and ocean forcing. Their use a numerical model to demonstrate that basal conditions are best approximated as Coulomb plastic and that other commonly used assumptions cannot explain the observations. The model is then used to predict how much ice will be lost to dynamic thinning in this rather large drainage basin over the coming decades. The predicted sea level rise contribution of 13-15 mm by 2100 is significantly more than previously predicted, owing to the slippery basal conditions.

Originality and significance:

The observed acceleration, accompanied by thinning, is clearly very significant. The mechanism, known as dynamic thinning, sustains sea level rise from Greenland and shows how the ice sheets interaction with the ocean extends far inland. While this has been known for some time in Antarctica, inland thinning to this extent has not previously been observed in Greenland, where ocean forcing are typically concentrated much closer to the coast. To my knowledge, the only previous observation of inland thinning was by Doyle et al. GRL 2013, and that was 90 km inland of the southwestern ice margin much farther south, which is land-terminating and the thinning was therefore not attributable to the ocean. Clearly this catchment has the potential to influence sea level rise to a major extent, as do specific catchments such as that of Thwaites in West Antarctica. The findings that are reported here are important. Greenland is often treated as a whole in sea level studies because it has so many fast flowing outlets, but this catchment stands out and deserves special attention.

Data & methodology:

The inland extend of acceleration and thinning is verified by multiple data sources and it is therefore a robust observation. While the original source may have been the ground-based GPS records, the authors have corroborated the recent increase in speed 90 - 190 km from the coast using satellite observations and airborne altimetry. The authors have gone to great extent to investigate dynamic thinning, not only by setting up and managing GPS stations in a very extremely remote location; observing relatively small-magnitude changes so far inland is also no easy task using satellite data. The authors should be commended for those efforts as well explaining what they have discovered quantitatively with a numerical model. The study is in my view very robust.

Appropriate use of statistics and treatment of uncertainties:

Uncertainties are clearly explained.

Conclusions:

The conclusion are robust, reliable and valid.

Suggested improvements:

I do have some queries about the GPS records which seem noisy to me. See technical comment #1 below. I also don't understand why the various data sources are presented with data extracted from seemingly different places in the catchment. Why not use the GPS locations? See technical comment #2 below. There are some central features of the numerical model, which can be better explained. See technical comment #3 below.

References: appropriate credit to previous work?

Yes, with the suggestion of also citing Doyle et al. GRL 2013 who reported inland thinning 90 km inland of the land ice margin near Russell Glacier in the southwest. The source of thinning there was surface melting and runoff, which is different; but I nonetheless think it would be valuable to

mention that study too.

Clarity and context:

The manuscript is short and succinct and well written. I do think there is room to expand a bit on some of the technical aspects, e.g. when addressing the technical comments below.

Technical comments:

Technical comment #1: there is quite a bit of variation in the daily mean velocity derived by GPS (Fig. 2a and Extended Fig. 4). While the linear increase over time is very clear - I'm not questioning that - it is difficult to understand why there is such scattered intra-seasonal variability. The spread from day to day is larger than what I would have expected given that the ice flow is quite fast. Is this related to GPS accuracy? Is the receiver L1 only? Gipsy is a good software package to use, but could the data be processed differently somehow? A few more details on the GPS in the relevant supplementary section would be good. Given that satellite data confirms the observed trend in the GPS records, it would not be a major concern if further processing of the GPS would not markedly improve the accuracy. Both speed up and thinning are unequivocal in all records.

Technical comment #2: It would make sense if the ice speed shown in Extended Fig. 5 were from the approximate same position as the GPS stations, i.e. as described in the section on correcting the GPS velocities. But that seems to not be the case for the data shown in the figures. Similar for Extended Fig. 6 and 7, which shows extracted values from Cryosat-2 and ICESat-2. These data are great, but it would make sense to link the various examples as far as possible.

Note to editor: GPS captures ice sheet motion in a Lagrangian reference frame, because the GPS stations advect with the ice. Satellite observations are Eulerian, which means time series capture data from geographical fixed points. The authors have made corrections that make it possible to compare the derived surface motion with the flow of ice observed at fixed locations by satellite. This is appropriate and detailed. It would be informative to compare data for the same fixed points, insofar as it is possible and makes sense to do so. It should be noted that comparing Lagrangian and Eulerian are difficult and that there are sound technical reasons for these data to not necessarily match up within the anticipated error of each technique.

Technical comment #3: Some key features of the model could be explained better. The value of the yield strength used with von Miss calving criteria should ideally be stated explicitly somewhere. It would also be nice to know what the plastic yield strength of the bed approximately is, insofar as the model informs it. There should also be brief summary of the frontal dynamics, e.g. what is the basal melt rate under the floating ice given the oceanic forcing? How far inland is the frontal retreat in the future scenarios? Finally, it should be noted that the basal shear strength will probably evolve over time, whereas the model assumes it is constant. Bougamont et al. JGR 2019 report weakened plastic bed beneath Pine Island Glacier in response to speed up in a similar manner in West Antarctica. The same could well happen here, which means future contributions to sea level rise could be even higher than estimated, with stability necessarily provided by the ice stream's shear margins (as shown by Bougamont et al. for PIG)

Comments to figures:

Figure 1 is probably a bit larger than it needs to be and the colour scale needs a bar.

Figure 2 could present the GPS data better. Panels a), b) and c) are very small. Can they be expanded? Can the data can be processed so that it scatters less (see technical comment #1 above).

Figure 3 should inform what the various data sources are, i.e. Cryosat-2 etc, etc.

Figure 4 should mention where the data are from (NGIS) and what “observed” refers to in terms of mass change.

All figures would benefit from slightly expanded captions.

Line by line comments:

L16-17. I recommend including specific mass loss estimates to be specific.

L20-21. Rather than catastrophic collapse, I recommend rapid retreat leading to unstable conditions similar to the marine based setting of ice streams in Antarctica. Maybe explain that NGIS is unique for its similarity to Antarctic ice streams and that it is the only glacier of this type in Greenland.

L25-26, “identify the correct basal conditions...”. I don’t think the basal conditions have been identified, but you are demonstrating that a plastic approximation is better than other solutions. Correct basal conditions, e.g. as identified in Antarctica by Kamb and the Caltech group in the 1990s, means observations to identify what the plastic yield strength is and how it changes according to water content, etc.

L63-64, sentence needs commas, otherwise it sounds as if frontal changes take place 190 km inland. Also, please explain “mirror”. Is the seasonal variations inland mirroring the seasonal variations at the coast?

L58-63. Explain why daily means are used to subsequently derive monthly means? Is the RMS of the latter (when calculated this way) the correct uncertainty?

L73. “...(see Extended Data Fig. 5)”. Caption of figure states: “Errorbars denote speed...”. Try to explain the error bar better. What exactly is the uncertainty?

L83. It is great that you are also able to show the thinning of the glacier so accurately. This additional information is very valuable.

L97-101. It’s important, but not surprising that regularised Coulomb friction law performs better than the other basal parameterisations given how far inland the acceleration extends. Somewhere, there needs to be a brief discussion of what makes the bed plastic. Coulomb friction also did well in simulations of the land-terminating margin by Bougamont et al. Nat Comms (2014). Shapero, Joughin et al. JGR (2016) inferred plastic beds beneath some of Greenland’s largest outlets from inversions (not incl. NEGIS). A summary of the previous work and whether plasticity arises from deformation of basal sediment or the sliding mechanism itself should ideally be included somewhere in the text.

L124-125. “All model simulations lead to a collapse..” It would be nice to show what exactly happens at the front, e.g. how far inland does the front of ZI retreat when you run the future simulations? What is the calving rate? These are all interesting quantities to report from the model, somewhere, if not in the text then in the supplementary.

L133. “ when the ice shelf collapsed...”. Did the whole ice shelf disappear? If not, re-word to parts of the ice shelf collapsed.

Extended data fig. 7 seems to have an error in the labelling.

Author Rebuttals to Initial Comments:

Referees' comments:

Referee #1 (Remarks to the Author):

A Summary of the key results

The manuscript conclusively shows that one of the major outlet glaciers of the Greenland Ice Sheet (GrIS) underwent very large dynamic and geometric changes in the last decade. These alarming changes will likely continue into the future, as predictions from an ice flow model show, that was calibrated with the presented measurements. This large and very remote sector of the GrIS will contribute considerably to global sea level rise.

Authors: Thank you very much for reviewing this paper. In the light of the insightful response from you, reviewers #2 and #3, we have made several changes, which have greatly improved the revised manuscript. A detailed response to your comments addressing all of the identified issues is listed below.

B Originality and significance: if not novel, please include reference

This study is a milestone in ice sheet research. It is a very important and comprehensive investigation on one of the crucial topics of glaciology, climate change research, and therefore for all of humanity. Rapid changes in major Greenland Ice Sheet drainage basins have world-wide consequences by iceberg production, altering ocean fresh-water fluxes, and obviously sea level rise.

The manuscript reports on a unique data set from field measurements from one of the most challenging places on the planet. In-situ ice flow velocities are a key quantity to understand rapid ice flow which dominates the short-term changes of polar ice sheets. The acceleration and rapid thinning of the huge NEGIS ice stream is unprecedented and of highest importance for the present and future evolution and stability of the ice sheet.

C Data & methodology: validity of approach, quality of data, quality of presentation

The data set is unique and hard-earned. Obtaining good in-situ data from the Greenland ice sheet is very difficult, and from an extremely remote area like North-East Greenland even more. A consequence of this are the large data gaps in winter. Nevertheless, the data analysis has been carefully done, is convincing, and provides new and unique results. These findings are corroborated by analysis of state-of-the-art satellite data products. The model results were obtained with one of the leading ice sheet models, and properly parametrized and validated to emulate past behavior. This is crucial for predicting the future evolution of this ice stream.

D Appropriate use of statistics and treatment of uncertainties

The quantification of uncertainties is reasonable, and the discussion of measurement errors is good

and sufficient.

E Conclusions: robustness, validity, reliability

The conclusions are very relevant, and undeniably robust. Using very different data acquisition systems to independently arrive at a set of consistent conclusions leaves little room for doubt of the robustness and validity of the conclusions. Obviously, the future evolution hinges on the accuracy of prescribed climate scenarios. The variability of the results by using different forcings were appropriately explored and quantified.

F Suggested improvements: experiments, data for possible revision

The manuscript should be streamlined by a very careful native speaker. Especially the last sections and the "Methods" need a lot more attention to precise and concise language.

Authors: The entire manuscript including the section "Methods" has been streamlined by a native speaker and professionally proofread.

More careful descriptions of the "models" in the main text, and especially in the "Methods", have to be given. It seems that the term "these models" (lines 104, 113, 128) refers not to different modeling codes (which I would call "models"), but merely to different subroutines/parametrizations for the calculation of basal friction within the same code base. It is crucial to very clearly state this.

Authors: we have decided to keep the model section short. It is important to note that our numerical ice flow model is identical to Choi et al. (2021). Extended Data Table 1 shows all model results. Here, a model is defined by choice of Friction law, RCP scenario, and GCM forcing.

In line 107 it should be stated what MAR does, rather what the acronym stands for (not all readers know French). Line 113 should start "The model results show..."

Authors: "Modèle Atmosphérique Régional" is replaced with "Regional Atmosphere Model". MAR is a surface mass balance model, net balance between the processes of accumulation and ablation. Line 113 was corrected accordingly.

I think there is an error on line 115: only 200 km² are affected? This cannot be true, it has to be 2-3 orders of magnitude more, otherwise there would be no noticeable effect on SLR. Also the maps in Fig. 3 show an area of about 200 x 100 km that is affected by thinning.

Authors: It is a misprint. 200 km² is replaced with 20,000 km².

Generally, the Figure style should be consistent, e.g. quantities plotted should be annotated by e.g. "Sea level change" (capitalized). At the moment this is random and inconsistent. Even if English has no reasonable rules for this, it should be consistent.

Authors: changed accordingly.

Figure captions should start capitalized (ATM this is not consistent).

Authors: changed accordingly.

Figure 1 is blurry. It has terrible colors for speed. A better colormap (like used in many other studies, e.g. Rignot), and a colorbar are necessary. Are the speeds shown on a linear or logarithmic color scale (for Greenland this is likely log)? Annotations NEG1-3 are barely readable, use better colors.

Authors: we have changed the figure accordingly. We have added a color bar. Ice speeds are shown on a logarithmic color scale.

Figure 2 is blurry. Use consistent label capitalization ("Ice speed"). Blue circles marking positions of NEG1-3 are barely visible on red, use better colors.

Authors: changed accordingly

Figure 3: 3k: capitalize "Cumulative". Captions starts with lower-case text. Why is this called "Cumulative thinning"? It is just thickness change. Easier to understand and more precise, too.

Authors: changed accordingly

Showing these rates in Figure 3, is it assumed that the rate of thinning is constant? Why not show thinning from year to year, but always with the 2011 baseline?

Authors: No, rate is not constant. We show thinning relative to 2011, because then it is easy to see the inland propagation of thinning. However, we do also show thinning from year to year (see Extended Data Fig 7.)

The crucial Figure 4 is blurry. Why is this called "Cumulative mass loss"? It is just mass change/loss. Easier to understand and more precise, too.

Authors: changed accordingly

I was quite surprised that not more care has been given to a manuscript submitted to a scientific journal. While the main text is well-written, the last section (dwellings on Jakobshavn Isbrae) seems to be a last-second addition without any streamlining or peer-review by co-authors. The figures look like a first draft, blurry, inconsistent, annotations that are difficult to read, bad color maps.

Authors: we have carefully updated all figures and text.

The supplementary material is full of typos (sometimes even funny), unclear formulations, and missing explanations on many aspects of the modeling exercise. All of this should be very carefully revised and clarified.

Authors: The numerical simulations are strictly identical to Choi et al 2021, except for basal sliding. Only friction laws are changed here. The supplementary material has been streamlined by a native speaker.

Line 92: Approximation of what, why higher order? (I understand it, but not everybody is ice sheet modeler). Better leave this away, or give a precise description. As is this is only confusing. Rather say FE, FD, FV, and say which equations are solved (likely a nonlinear diffusion equation?)

Authors: We agree that this is a technical detail and removed it from the "Numerical ice flow model"

section (main text), and only kept it in the Methods section with a clear reference to the paper that describes this simplified momentum balance:

Pattyn, F., A new three-dimensional higher-order thermomechanical ice sheet model: Basic sensitivity, ice stream development, and ice flow across subglacial lakes, *J. Geophys. Res.*,108(B8), 2382, doi:10.1029/2002JB002329, 2003.

Line 59: What is the width of the stream? How representative are GPS velocities?

Authors: The stream is about 50 km wide (see the fast-flowing part in figure 1). We added the glacier length and width to line 38. GPS are point measurements but represent a wide area as shown in fig 2d (acceleration from Sentinel-1).

Line 77: Explain what you mean by this. Mass continuity requires that, extension => thinning.

Authors: Previous work (e.g. King et al., 2020) has shown glacier retreat causes dynamic thinning.

King, M.D. et al. Dynamic ice loss from the Greenland Ice Sheet driven by sustained glacier retreat. *Commun. Earth Environ.* **1**, 1 (2020).

Line 82: "Land" capitalize like the rest

Authors: corrected

Line 86: Is this only dynamic thinning? What is the contribution of surface melt?

Authors: we do not separate SMB and dynamic induced thinning in this study. We simply focus on the total mass loss.

Line 91: the word "model" in "linear model" means something completely different from later uses. So better call this "linear fit/approximation/etc"

Authors: we assume you refer to line 60. Corrected.

Line 113: "The model results show..."

Authors: corrected

Line 131: from the 1970s to ...

Authors: corrected

Line 138: wrong plural "an ice sheets "

Authors: corrected.

Line 151: "sea level rise estimates."

Authors: changed.

Line 181: "de-acceleration" could be "slowdown"

Authors: corrected

G References: appropriate credit to previous work?

Previous work has been appropriately given credit. But the bibliography itself is an assembly of random styles and typesetting, DOIs are mostly missing, several typos in journal names and titles. Again, surprising that this was never peer-reviewed by the co-authors.

Authors: We have carefully corrected the bibliography and used nature style. Nature does not use DOIs.

H Clarity and context: lucidity of abstract/summary, appropriateness of abstract, introduction and conclusions

Abstract and conclusion are clear, appropriate and well-written. The introduction is easy to understand and well crafted. The conclusions could be somewhat sharpened and streamlined.

Authors: Once again thanks for reviewing this paper.

Referee #2 (Remarks to the Author):

Review of:

Extensive inland thinning and speed-up of the Northeast Greenland Ice Stream

By Khan et al. for Nature (Manuscript ID: 2022-05-08091)

General Comment

Even though we are living in the era of satellites, ground observation provides unique data. However, both ground and satellite observations cannot look into the future evolution of glaciers, while simulations can do. Khan and coauthors combine the GPS observation, satellite observation and simulations to investigate the past and future changes of the Northeast Greenland Ice Stream, where the largest basin of Greenland locates. The data are valuable and the methodology is feasible. This work is very interesting not only to glaciologists but also to the public due to its potential contribution to global sea-level rise. I find the idea, goal and outcome of the study – the first acquisition of a set of in-situ ice flow velocity measurements and subsequent simulations on ice stream dynamics in time and space – in principal worth publishing. However, I do think that the authors could do a better job in explaining the value of their research.

Therefore, I would highly recommend a revision of the manuscript taking the following into consideration.

Authors: Thank you very much for reviewing this paper. In the light of the insightful response from you, reviewers #1 and #3, we have made several changes, which have greatly improved the revised

manuscript. A detailed response to your comments addressing all of the identified issues is listed below.

Firstly, the manuscript has presented the acceleration of surface ice flow speed, however the speed itself is not presented along the simulations from 2020 to 2100, while the ice flow speed is crucial to the ice flux computation, which is related to the mass balance of glaciers.

Authors: correct, we focus on ice flow acceleration and mass change in this paper. To initialize the flow model, we used observed 2007–2008 surface velocities (see Choi et al. (2021)). After 2007 the modelled speed can change freely and is not constrained to observations. Therefore, it makes sense to show changes in speed rather than the speed itself since the initial state of the model coincides with 2007/2008 measured speed.

Secondly, the in-situ GPS observations are valuable. What's the type of GPS receiver/antenna for the 3 GPS stations? It's a pity that the GPS observations were not continuously from 2016 to 2020, why there are so many interruptions? How to get the observed accelerations in the missing periods? It seems that a precise point positioning method was used to analyze the GPS data using GIPSY-OASIS, is it possible to get more precise positioning results using differential GPS method? Why using monthly averaged velocities rather than daily or weekly ones? Furthermore, there is no indication for the GPS data in "Data availability"?

Authors: We used Javad Delta-3 GNSS receivers and Javad GrAnt-G3T GNSS Antennas. The data is not continuous because we used solar panels to recharge batteries. We tried with a wind turbine at NEG1, but is still difficult to obtain continuous data. There can be large periods with no wind and during the winter months (November-February) there is no sunlight to recharge batteries. We likely need a larger battery pack to retrieve year-round continuous data. Another problem is that the temperature goes down to -50 or -60 °C during winter. This is not good for the batteries or the GNSS receiver. This causes huge problems and sometimes the receiver does not start up automatically when the sunlight comes back in spring.

We use monthly means to take temporally correlated errors into account (see Khan et al., 2007). Khan, S. A., Wahr, J., Stearns, L. A., Hamilton, G. S., van Dam, T., Larson, K. M., and Francis, O.: Elastic uplift in southeast Greenland due to rapid ice mass loss, *Geophys. Res. Lett.*, 34, L21701, <https://doi.org/10.1029/2007gl031468>, 2007.

We estimate a mean acceleration during the entire observational period. We do not need to estimate separate accelerations at the missing periods.

Note, we estimate positions for every 15 sec. see Extended Data Fig. 2. For a station near the margin of 79 glacier, we observe tides. However, data from this station was recently published by Christmann et al., 2021. The study by Christmann et al. (2021) compared GIPSY solutions with the Waypoint GravNav 8.8 processing software and obtained very similar results.

Christmann, J., V Helm, S A Khan, T Kleiner, R Müller, M Morlighem, N Neckel, M Rückamp, D Steinhage, O Zeising, and A Humbert, Elastic deformation plays a non-negligible role in Greenland's outlet glacier flow, *Commun Earth Environ*, 2, 232, <https://doi.org/10.1038/s43247-021-00296-3>, 2021.

Finally, GPS data is available and this is stated in the “Data availability” section. [GNSS daily solutions are available at https://datadryad.org/stash/share/VxgLxKo7i4_u8Fy8p4CGj37PxASlfc9hiAGpDgiHzQM](https://datadryad.org/stash/share/VxgLxKo7i4_u8Fy8p4CGj37PxASlfc9hiAGpDgiHzQM)

Thirdly, the propagation of inland thinning would, as the authors described, increase the future contribution of Greenland to sea-level rise and reduce the uncertainty of simulation. But, how the propagation of inland thinning itself will change the basal conditions of the ice stream in the future? If the inland thinning did change the basal conditions, would it increase the uncertainty of modelling? Glacier flow speed-up is common around the Arctic (<https://doi.org/10.1007/s13131-021-1718-1>), then is it possible to quantitatively analyze the factors behind recent ice flow speed-up of the Northeast Greenland Ice Stream?

Authors: We use the GPS time series to see which friction law provides the best match to ice flow speed. All sliding laws vary in time, as a function of basal sliding speed and/or ice thickness (overburden pressure), etc. We do, however, keep the friction coefficient constant in time after calibration in 2007. It could potentially change further if the subglacial hydrological system changes significantly, but this is beyond the scope of this work. We added this point in the main text.

There are also many issues that should be fixed throughout the manuscript. More details of comments are as follows.

Specific Comments

*Page 2 Line 35:

"GIS" is widely used for "Geographical Information System". I have a suggestion that "Greenland Ice Sheet (GIS)" be tuned into "Greenland Ice Sheet (GrIS)". And this change will impact the whole manuscript.

Authors: replaced accordingly.

*Page 2 Line 46:

"retreat of the ice front" is not an certain reason of "rapidly speeding up", and the "rapidly speeding up" is not the inevitable result of glacier "retreat of the ice front". This sentence should be revised.

Authors: we have deleted "...due to the retreat of the ice front of ZI"

*Page 3 Line 108:

I suggest that "GCM" should be explained with a full name or references.

Authors: "*General Circulation Models*" added

*Page 3 Line 114:

I think "GIS" here should be "NEGIS", because the results shown in the paragraph are limited in the NEGIS.

Authors: corrected.

*Page 4 Line 117:

It is Fig. 3I that displays the observed thinning rate during the last decade, not Fig. 3m.

Authors: corrected.

*Page 4 Line 138:

Repeated: "in the in the".

Authors: corrected

*Page 4 Line 151:

"reduce uncertainty in of future sea level rise" should be checked.

Authors: corrected

*Page 8 Fig.4:

All the "RPC" in the legend should be replaced with "RCP".

Why the Figure 4 only indicates error bars for the Observed curve, no error for the simulation results?

Authors: corrected. simulations do not include error bar. Instead, an ensemble of simulations is shown (which is fairly common in climate modeling).

Appendix:

*Page 14 Line 379:

"august" should be "August".

Authors: corrected

*Page 18 Line 456:

In the caption of "Extended Data Fig. 7", the "(c) CryoSat-2 elevations at P2" should be "(c) CryoSat-2 elevations at P1"

Authors: corrected

*Page 22 Line 534:

The "of" in "GNET-GIA empirical model of" is needless.

Authors: corrected

*Page 27 Line 620:

In Extended Data Table 1, "RPC" should be "RCP".

Authors: corrected

Referee #3 (Remarks to the Author):

Summary of the key results:

The authors present GPS records from the northeast Greenland ice stream, which show acceleration of 2.7 - 4.9 m/yr per year 90 - 190 km inland from the coast over a three year period. The GPS observations are corroborated by satellite observations, which include both speed and height changes in the northeast Greenland ice stream's catchment. The authors attribute dynamic thinning to ice shelf disintegration and ocean forcing. Their use a numerical model to demonstrate that basal conditions are best approximated as Coulomb plastic and that other commonly used assumptions cannot explain the observations. The model is then used to predict how much ice will be lost to dynamic thinning in this rather large drainage basin over the coming decades. The predicted sea level rise contribution of 13-15 mm by 2100 is significantly more than previously predicted, owing to the slippery basal conditions.

Authors: Thank you very much for reviewing this paper. In the light of the insightful response from you, reviewers #1 and #2, we have made several changes, which have greatly improved the revised manuscript. A detailed response to your comments addressing all of the identified issues is listed below.

Originality and significance:

The observed acceleration, accompanied by thinning, is clearly very significant. The mechanism,

known as dynamic thinning, sustains sea level rise from Greenland and shows how the ice sheets interaction with the ocean extends far inland. While this has been known for some time in Antarctica, inland thinning to this extent has not previously been observed in Greenland, where ocean forcing are typically concentrated much closer to the coast. To my knowledge, the only previous observation of inland thinning was by Doyle et al. GRL 2013, and that was 90 km inland of the southwestern ice margin much farther south, which is land-terminating and the thinning was therefore not attributable to the ocean. Clearly this catchment has the potential to influence sea level rise to a major extent, as do specific catchments such as that of Thwaites in West Antarctica. The findings that are reported here are important. Greenland is often treated as a whole in sea level studies because it has so many fast flowing outlets, but this catchment stands out and deserves special attention.

Data & methodology:

The inland extend of acceleration and thinning is verified by multiple data sources and it is therefore a robust observation. While the original source may have been the ground-based GPS records, the authors have corroborated the recent increase in speed 90 - 190 km from the coast using satellite observations and airborne altimetry. The authors have gone to great extent to investigate dynamic thinning, not only by setting up and managing GPS stations in a very extremely remote location; observing relatively small-magnitude changes so far inland is also no easy task using satellite data. The authors should be commended for those efforts as well explaining what they have discovered quantitatively with a numerical model. The study is in my view very robust.

Appropriate use of statistics and treatment of uncertainties:
Uncertainties are clearly explained.

Conclusions:

The conclusion are robust, reliable and valid.

Suggested improvements:

I do have some queries about the GPS records which seem noisy to me. See technical comment #1 below. I also don't understand why the various data sources are presented with data extracted from seemingly different places in the catchment. Why not use the GPS locations? See technical comment #2 below. There are some central features of the numerical model, which can be better explained. See technical comment #3 below.

References: appropriate credit to previous work?

Yes, with the suggestion of also citing Doyle et al. GRL 2013 who reported inland thinning 90 km inland of the land ice margin near Russell Glacier in the southwest. The source of thinning there was surface melting and runoff, which is different; but I nonetheless think it would be valuable to mention that study too.

Clarity and context:

The manuscript is short and succinct and well written. I do think there is room to expand a bit on some of the technical aspects, e.g. when addressing the technical comments below.

Technical comments:

Technical comment #1: there is quite a bit of variation in the daily mean velocity derived by GPS (Fig. 2a and Extended Fig. 4). While the linear increase over time is very clear – I'm not questioning that - it is difficult to understand why there is such scattered intra-seasonal variability. The spread from day to day is larger than what I would have expected given that the ice flow is quite fast. Is this related to GPS accuracy? Is the receiver L1 only? Gipsy is a good software package to use, but could the data be processed differently somehow? A few more details on the GPS in the relevant supplementary section would be good. Given that satellite data confirms the observed trend in the GPS records, it would not be a major concern if further processing of the GPS would not markedly improve the accuracy. Both speed up and thinning are unequivocal in all records.

Authors: variation in the daily mean velocity is partly due to gaps. We estimate mean velocity over 24 hours, however, for some days we have observations over only 12 or less. This causes scattered intra-seasonal variability. Another source of this scatter is summer speedups. The figure below shows the mean surface speed of about 20 km from the ZI terminus. There is a large speedup in May and a slowdown in July. This speedup likely propagates upstream and is captured by GPS measurement. However, the focus of this paper is long-term changes and not day-to-day variations.

In general, the spread from day to day is not related to GPS accuracy. We use Javad Delta-3 GNSS receivers and Javad GrAnt-G3T GNSS Antennas that record both L1 and L2 frequencies. See above comment to reviewer 2 regarding data processing.

Technical comment #2: It would make sense if the ice speed shown in Extended Fig. 5 were from the approximate same position as the GPS stations, i.e. as described in the section on correcting the GPS velocities. But that seems to not be the case for the data shown in the figures. Similar for Extended Fig. 6 and 7, which shows extracted values from Cryosat-2 and ICESat-2. These data are great, but it would make sense to link the various examples as far as possible.

Authors: changed accordingly. We have replaced fig 5b-d with time series of ice speed at NEG1, NEG2, and NEG3. However, for figs 6 and 7 we prefer to show time series near the margin and inland (all GPS stations are located inland).

Note to editor: GPS captures ice sheet motion in a Lagrangian reference frame, because the GPS

stations advect with the ice. Satellite observations are Eulerian, which means time series capture data from geographical fixed points. The authors have made corrections that make it possible to compare the derived surface motion with the flow of ice observed at fixed locations by satellite. This is appropriate and detailed. It would be informative to compare data for the same fixed points, insofar as it is possible and makes sense to do so. It should be noted that comparing Lagrangian and Eulerian are difficult and that there are sound technical reasons for these data to not necessarily match up within the anticipated error of each technique.

Technical comment #3: Some key features of the model could be explained better. The value of the yield strength used with von Miss calving criteria should ideally be stated explicitly somewhere. It would also be nice to know what the plastic yield strength of the bed approximately is, insofar as the model informs it. There should also be brief summary of the frontal dynamics, e.g. what is the basal melt rate under the floating ice given the oceanic forcing? How far inland is the frontal retreat in the future scenarios? Finally, it should be noted that the basal shear strength will probably evolve over time, whereas the model assumes it is constant. Bougamont et al. JGR 2019 report weakened plastic bed beneath Pine Island Glacier in respond to speed up in a similar manner in West Antarctica. The same could well happen here, which means future contributions to sea level rise could be even higher than estimated, with stability necessarily provided by the ice stream's shear margins (as shown by Bougamont et al. for PIG)

Authors: We added some details of the model results. As the reviewer mentioned, we assume that the friction coefficient is kept constant for future simulations. We also added this point in the main text.

Comments to figures:

Figure 1 is probably a bit larger than it needs to be and the colour scale needs a bar.

Authors: Figure 1 has been revised and colour scales were added.

Figure 2 could present the GPS data better. Panels a), b) and c) are very small. Can they be expanded? Can the data can be processed so that it scatters less (see technical comment #1 above).

Authors: scatters can not be reduced. See our response above. We have revised fig 2. Panels a), b) and c) are small here but also shown in full size in Extended Data Fig. 4.

Figure 3 should inform what the various data sources are, i.e. Cryosat-2 etc, etc.

Authors: Thinning is based on airborne and satellite altimetry. This is now stated in the figure caption title. The main text informs what data is used (ICESat-2, Cryosat-2, NASAs ATM).

Figure 4 should mention where the data are from (NGIS) and what "observed" refers to in terms of mass change.

Authors: observed refers to airborne and satellite altimetry. This is now stated in the figure caption.

All figures would benefit from slightly expanded captions.

Line by line comments:

L16-17. I recommend including specific mass loss estimates to be specific.

Authors: well, the abstract has to be very short. We prefer to discuss numbers in the main text.

L20-21. Rather than catastrophic collapse, I recommend rapid retreat leading to unstable conditions similar to the marine based setting of ice streams in Antarctica. Maybe explain that NGIS is unique for its similarity to Antarctic ice streams and that it is the only glacier of this type in Greenland.

Authors: the abstract has to fulfill nature requirements and be short and precise. We have added a sentence about ice streams in Antarctica.

L25-26, "identify the correct basal conditions...". I don't think the basal conditions have been identified, but you are demonstrating that a plastic approximation is better than other solutions. Correct basal conditions, e.g. as identified in Antarctica by Kamb and the Caltech group in the 1990s, means observations to identify what the plastic yield strength is and how it changes according to water content, etc.

Authors: agree, we have replaced "identify" with "select".

L63-64, sentence needs commas, otherwise it sounds as if frontal changes take place 190 km inland. Also, please explain "mirror". Is the seasonal variations inland mirroring the seasonal variations at the coast?

Authors: we have changed the sentence. Regarding your question. Yes, we do see summer speedup (May to July) at all GPS stations. However, we decided to keep this paper simple and not discuss the propagation of summer speedups here.

L58-63. Explain why daily means are used to subsequently derive monthly means? Is the RMS of the latter (when calculated this way) the correct uncertainty?

Authors: yes, correct. The GPS time series have error sources that produce temporal correlations. To take this temporally correlated (non-Gaussian) noise into account, we use the daily solutions for each station and estimate the monthly mean ice speed and associated errors.

L73. "... (see Extended Data Fig. 5)". Caption of figure states: "Errorbars denote speed...". Try to explain the error bar better. What exactly is the uncertainty?

Authors: The error bars denote the mean speed and standard deviation of the underlying shift maps generated by the offset tracking. The error bars are adopted from Solgaard et al. (2021).

L83. It is great that you are also able to show the thinning of the glacier so accurately. This additional information is very valuable.

Authors: thanks

L97-101. It's important, but not surprising that regularised Coulomb friction law performs better than the other basal parameterisations given how far inland the acceleration extends. Somewhere, there needs to be a brief discussion of what makes the bed plastic. Coulomb friction also did well in simulations of the land-terminating margin by Bougamont et al. Nat Comms (2014). Shapero,

Joughin et al. JGR O(2016) inferred plastic beds beneath some of Greenland's largest outlets from inversions (not incl. NEGIS). A summary of the previous work and whether plasticity arises from deformation of basal sediment or the sliding mechanism itself should ideally be included somewhere in the text.

Authors: We added the previous work and references in the main text.

L124-125. "All model simulations lead to a collapse.." It would be nice to show what exactly happens at the front, e.g. how far inland does the front of ZI retreat when you run the future simulations? What is the calving rate? These are all interesting quantities to report from the model, somewhere, if not in the text then in the supplementary.

Authors: The ZI front retreat additional 30 km inland. Rather than a figure, we have added an animation that shows the retreat using Regularized Coulomb friction law, with NorESM1, and RCP4.5.

L133. " when the ice shelf collapsed...". Did the whole ice shelf disappear? If not, re-word to parts of the ice shelf collapsed.

Authors: yes

Extended data fig. 7 seems to have an error in the labelling.

Authors: corrected.

Reviewer Reports on the First Revision:

Referees' comments:

Referee #1 (Remarks to the Author):

This is the second round of reviews for this manuscript. This means that points A to E and G/H are the same as in the last round, and are not repeated here.

The authors have done a good job taking into account all comments. They clarified and streamlined the manuscript. They also greatly improved the figures.

My recommendation at this point is to publish the manuscript, after the minor comments below have been taken into account.

F Suggested improvements: experiments, data for possible revision

The Figures were not labeled in the provided text, and especially the "extended Figures" were difficult to pin down correctly.

1 and many other places: why is "North" uppercase but "East" lowercase (are there any sensible capitalization rules in English)?

30 "higher-end projections": I have no idea what this means. Why not just say "projections"

50 Often more than just GPS (= the original US system) is used, and this is now termed GNSS (including GLONASS, BAIDU, GALILEO).

Starting from line 151 onwards (future observations) the term GNSS should be used in any case.

112 "te" -> "the"

159 Although Jakobshavn features a different bathymetry (very deep, narrow trough) and always flowed one order of magnitude faster than ZI.

168 "consist" -> "consists"

170 "Image" -> lowercase.

171 say "the satellite-derived surface speed" or "the surface speed derived from Sentinel-1 data"

175 (c) is "NEG1" (ERROR)

188 I still do not understand why "cumulative thinning" is used. This is just "thinning", or better "surface elevation change" (since we do not know what is happening under the ice, sediment erosion is potentially large).

197 "y-axis" -> "vertical axis". Why is the right axis green? I think black is better visible, and it does not relate to any green line in the plot.

329 links are bleeding into the next words

Table 1: The title of the last column should be as short as the content of the lines below: e.g. "SLR (mm)"

Referee #2 (Remarks to the Author):

Please see attached PDF.

Referee #3 (Remarks to the Author):

The authors have adequately addressed the queries raised in peer review. They have improved the clarity of the analysis and the quality of the figures is better. The technical queries I addressed are resolved and I have no further comments to make. I support publication of the manuscript in its revised form. The result is clear and clearly presented. The implications are important for our understanding of sea level rise from Greenland and how to address it with models.

Author Rebuttals to First Revision:

Referees' comments:

Referee #1 (Remarks to the Author):

This is the second round of reviews for this manuscript. This means that points A to E and G/H are the same as in the last round, and are not repeated here.

The authors have done a good job taking into account all comments. They clarified and streamlined the manuscript. They also greatly improved the figures.

My recommendation at this point is to publish the manuscript, after the minor comments below have been taken into account.

Authors: Once again, thank you very much for reviewing this paper. In the light of the insightful response from you and the other reviewers, we have made the suggested changes. A detailed response to your comments addressing all of the identified issues is listed below.

F Suggested improvements: experiments, data for possible revision

The Figures were not labeled in the provided text, and especially the "extended Figures" were difficult to pin down correctly.

Authors: we will make sure they are labeled before publication of the manuscript..

1 and many other places: why is "North" uppercase but "East" lowercase (are there any sensible capitalization rules in English)?

Authors: corrected to North-East.

30 "higher-end projections": I have no idea what this means. Why not just say "projections"

Authors: "higher-end projections" is important here. The sea level rise of 13.5 to 15.5 m belongs to higher-end projections, compared to earlier study that suggest 1.5 to 3.3 mm, which belongs to "lower-end projections".

50 Often more than just GPS (= the original US system) is used, and this is now termed GNSS (including GLONASS, BAIDU, GALILEO).

Authors: Agree, we have replaced "GPS" with "GNSS" throughout the manuscript.

Starting from line 151 onwards (future observations) the term GNSS should be used in any case.

112 "te" -> "the"

Authors: corrected

159 Although Jakobshavn features a different bathymetry (very deep, narrow trough) and always flowed one order of magnitude faster than ZI.

Authors: we agree that Jakobshavn features a different bathymetry. But GNSS data are limited to lower sectors (0 to 100 km from the margin). Inland data (>100 km) could potentially improve projections.

168 "consist" -> "consists"

Authors: corrected

170 "Image" -> lowercase.

Authors: corrected

171 say "the satellite-derived surface speed" or "the surface speed derived from Sentinel-1 data"

Authors: done

175 (c) is "NEG1" (ERROR)

Authors: corrected

188 I still do not understand why "cumulative thinning" is used. This is just "thinning", or better "surface elevation change" (since we do not know what is happening under the ice, sediment erosion is potentially large).

Authors: we agree. changed to "thinning".

197 "y-axis" -> "vertical axis". Why is the right axis green? I think black is better visible, and it does not relate to any green line in the plot.

Authors: changed accordingly.

329 links are bleeding into the next words

Authors: corrected.

Table 1: The title of the last column should be as short as the content of the lines below: e.g. "SLR (mm)"

Authors: changed accordingly.

Referee #2 (Remarks to the Author):

A. Summary of the key results

The manuscript represents the crucial function of in-situ measurements together with satellite observations, which makes a key factor in modelling the future glacier evolution. As a rareknown part of Greenland, the Northeast Greenland Ice Stream has its own dynamic process, and this manuscript revealed some features behind, showing the extensive inland thinning and speed-up of that Stream.

I'm pleased to see the revised manuscript, which has been improved greatly. And I suggest that the manuscript be accepted after addressing some minor issues below.

Authors: Once again, thank you very much for reviewing this paper. In the light of the insightful response from you and the other reviewers, we have made the suggested changes. A detailed response to your comments addressing all of the identified issues is listed below.

B. Originality and significance: if not novel, please include reference

The manuscript is original and the research work is significant in my opinion.

C. Data & methodology: validity of approach, quality of data, quality of presentation

The data set, especially the in-situ GPS data, is hard-earned. Although the time series of the GPS is not always continuous, the data set is unique and fill the blank in this remote field area. The only issue I found is that the GNSS daily solutions of NEG1 and NEG2 had some jump positions (as shown in figures below, derived from the data downloaded from https://datadryad.org/stash/share/VxgLxKo7i4_u8Fy8p4CGj37PxASIfC9hiAGpDgiHzQM).

How did the authors handle the jump positions in 2018?

Authors: Thanks a lot for looking into this. Outliers were unfortunately not removed in this version of the uploaded data files. We have uploaded new files where outliers are removed.

To remove outliers, we fit and remove a trend to each time series of speed, latitude and longitude. We estimate the mean of detrended speed, latitude and longitude.

We define outliers as value greater than 3 Standard Deviations from the Mean.

For NEG1 we removed in total 8 data points or $(8/387=0.02)$ 2% of data.

For NEG2 we removed in total 4 data points or $(4/195=0.02)$ 2% of data.

D. Appropriate use of statistics and treatment of uncertainties

The treatment of uncertainties is reasonable.

E. Conclusions: robustness, validity, reliability

The conclusions are robust and reliable. The only issue is in Line 155: “Jakobshavn Isbræ (JI) drains 6% of the GrIS”. Here the 6% has more than one interpretation: 6% of the watershed area of GrIS, or 6% of the whole mass of GrIS, or 6% of the entire runoff of GrIS. In order to avoid ambiguity, the authors should clearly indicate which one is the right meaning here. Although careful readers can get the answer after looking into the citations.

Authors: we mean drainage area of the GrIS.

F. Suggested improvements: experiments, data for possible revision

Parameterization is the key part of ice flow modelling. As I indicated in round one review, how the propagation of inland thinning itself will change the basal conditions of the ice stream in the future? Since the authors calibrated the friction coefficient in 2007, they can also test or calibrate the coefficient in a future time node, such as 2057 or 2100, with changed relevant ice flow velocity, ice thickness, etc. However, the evolution on subglacial hydrological system is complex, it may not be necessary to solve this issue in this manuscript. I hope that the modelling with changing basal conditions of Northeast Greenland Ice Stream could be addressed in their future studies.

Authors: we can only calibrate the friction coefficient over the observational period, so we cannot calibrate it in 2057 or 2100. The fact that the modeled velocity, calibrated in 2007 only, remains in excellent agreement with the observations until 2020 suggests that the parameterization is capable of capturing the changes in basal conditions as the ice accelerates and thins. It is true that we do not include changes in subglacial hydrology (which probably have not changed significantly over the past 10+ years, but may change over the course of the century). Subglacial hydrology remains an active area of research and how more meltwater will affect ice stream dynamics remains an open question that will hopefully be addressed in future studies.

G. References: appropriate credit to previous work?

Previous work has been credited appropriately.

H. Clarity and context: lucidity of abstract/summary, appropriateness of abstract,

introduction and conclusions

The manuscript is well written now

Referee #3 (Remarks to the Author):

The authors have adequately addressed the queries raised in peer review. They have improved the clarity of the analysis and the quality of the figures is better. The technical queries I addressed are resolved and I have no further comments to make. I support publication of the manuscript in its revised form. The result is clear and clearly presented. The implications are important for our understanding of sea level rise from Greenland and how to address it with models.

Authors: Great, Thanks!